# Computing time-dependent activity rate using non-declustered and declustered catalogues. A first step towards time-dependent seismic hazard calculations for operational earthquake forecasting

David Montiel-López[1], Sergio Molina[1, 2], Juan José Galiana-Merino[3, 4], Igor Gómez[1, 2], Alireza Kharazian[1], Juan Luis Soler-Llorens[5], José Antonio Huesca-Tortosa[6], Arianna Guardiola-Villora[7], and Gonzalo Ortuño-Sáez[8]

[1]Multidisciplinary Institute for Environmental Studies "Ramón Margalef" (IMEM), University of Alicante, Ctra. San Vicente del Raspeig, s/n, 03080 Alicante, Spain

[2]Department of Applied Physics, University of Alicante, Ctra. San Vicente del Raspeig, s/n, 03080 Alicante, Spain

[3]University Institute of Physics Applied to Sciences and Technologies, University of Alicante, Ctra. San Vicente del Raspeig, s/n, 03080 Alicante, Spain

[4]Department of Physics, Systems Engineering and Signal Theory, University of Alicante, Ctra. San Vicente del Raspeig, s/n, 03080 Alicante, Spain

[5]Department of Earth and Environmental Sciences, University of Alicante, Ctra. San Vicente del Raspeig, s/n, 03080 Alicante, Spain

[6]Department of Architectural Constructions, University of Alicante, Ctra. San Vicente del Raspeig, s/n, 03080 Alicante, Spain

[7]Department of Continuum Mechanics and Theory of Structures, Universitat Politècnica de València, Camino de Vera, s/n, 46022 Valencia, Spain

[8]Municipality of Orihuela, 03300 Orihuela, Spain

**Correspondence:** David Montiel-López (david.montlop@ua.es)

**Abstract.** Probabilistic Seismic Hazard Analysis (PSHA) typically requires tectonic b-values and seismic activity rates using declustered catalogues to compute the annual probability of exceedance of a given ground motion (for example, the peak ground acceleration or PGA). In this work, we propose a methodology that includes the spatially-gridded time-dependent b-value and activity rate computation using seismic clusters in PSHA calculations. To account for the the spatial variability and

the relationship of the earthquakes with the seismic sources, we incorporate the distance from the centre of the grid cell to the closest fault and the epicentre's uncertainty into the smoothing kernel as the average distance and the variance, respectively. To illustrate this methodology, we selected two scenarios as representatives of high-seismicity region and low-to-moderate seismicity region. The first one located in central Italy where L'Aquila earthquake happened while the other is in South-eastern Spain, where several earthquakes with a moment magnitude (Mw) greater than 4.0 have taken place over the last 30 years,

including two earthquakes with magnitude greater than or equal to 5.0. We compared three different seismic activity models based on the parameters considered in the calculations (distance from spatial cells to faults and epicentral distance uncertainty) and we defined and calculated the changes of the annual probability of exceedance for a given background PGA value. The results reveal noticeable changes of the annual probability of exceedance in the proximity of the occurrence of significant events. In the case of Italy, the annual probability of exceedance increases significantly, but in the case of Spain not all the

earthquakes have an associated increase in the exceedance probability. However, we have observed how, for moderate to low

seismicity regions, the use of a non-declustered catalogue can be appropriate when computing time-dependent PSHA, as in the case of Spain.

## 1 Introduction

Probabilistic Seismic Hazard Analysis (PSHA) has been the basis for seismic engineering design since Cornell (1968) proposed

it in order to account for all the possible earthquake scenarios and ground motion levels that can occur in the different seismic sources affecting the site of interest. One of the key points of PSHA is how the uncertainties are incorporated into the ground-motion computation, so the results are much more appropriate for use in engineering decision-making for risk reduction. However, the procedure increases in complexity (Budnitz et al., 1997).

PSHA results will depend on combining the pertinent input models (those which, according to the scientific and engineering

communities, represent the relevant phenomena in an appropriate way). Therefore, the choice of these models will evolve as our knowledge of the seismic activity and occurrence increases.

PSHA determines the probability of exceeding the ground motion level over a specified time period based on the occurrence rate of earthquakes and Ground Motion Prediction Equations (GMPEs). The occurrence rate of earthquakes is generally described by the truncated exponential model (Cosentino et al., 1977) and the characteristic earthquake model (Schwartz and

Coppersmith, 1984). Additionally, this earthquake occurrence rate or activity rate is assumed constant during the computation process. Therefore, it provides results which can be used for the seismic design. Once the knowledge of the seismic activity and occurrence improves due to the recording of new rare events or new tectonic information and models, the PSHA can be calculated again, and the seismic building codes will be updated if needed.

On the other hand, many authors have begun to focus the PSHA computations from a temporal or 'real-time' perspective, so

the term 'time-dependent probabilistic seismic hazard – TDPSHA' is now widely used. They are based on how the probabilities of large events increase as stress builds up on a fault plane until it reaches the breaking strength of the rock (Kanamori and Brodsky, 2004) and also how the probabilities of large aftershocks are a decreasing function after the main large event (Ogata, 1988; Reasenberg and Jones, 1989). However, measuring changes in the stress caused by the main shock is possible only indirectly and with somewhat low precision.

In general, small earthquakes are more frequent than large earthquakes. This is quantitatively stated in the Gutenberg–Richter law (Gutenberg and Ritcher, 1956) (G-R from now on) that can be seen in Eq. 1:

$$\log_{10} N\left(M \geq m\right) = a - b \cdot m \tag{1}$$

where $N$ is the cumulative number of earthquakes with magnitude $M$ above $m$, the $b$ value is the average size distribution of earthquakes (which expresses the ratio between high magnitude and low magnitude earthquakes) and $a$ is the productivity, or

more precisely, $10^a$ is known as the seismic activity rate. As the PSHA results are given as an annual probability of exceedance for a given intensity of the ground motion, the most common way to work with the activity rate is using the annual activity rate, which is obtained by dividing $10^a$ by the duration in years of the seismic catalogue. So, if we can identify seismic sources

a priori, then the seismic data inside each seismic source is used to compute a source-specific ($a$ and $b$) magnitude frequency distribution. However, it is often challenging to identify the corresponding boundaries and to have enough data allowing a significant statistical fitting.

Therefore, Frankel (1995) instead of specifying spatial borders for each seismic source adopted a boundary-less source model when computing the PSHA for central and eastern United States. Under this approach, the historical seismicity is spatially smoothed, and activity rates are computed at a grid of locations through the analysis domain. First, he divided the region into a grid, and then he counted for each cell of the spatial grid the number of earthquakes greater than a reference magnitude ($M_{ref}$) depending on the occurrence year of the event (1700 for magnitudes greater than 5.0 Mw and 1924 for magnitudes greater than 3.0 Mw). Next, the author obtained a maximum likelihood estimate for $10^a$ (Weichert, 1980) that he would then smooth using a Gaussian kernel with a correlation distance, $c$, of 50 km. This normalised smoothed value, $\tilde{n}_i$, was calculated as follows (Eq. 2):

$$\tilde{n}_i = \frac{\sum_j n_j \cdot \exp(-\Delta_{ij}^2/c^2)}{\sum_j \exp(-\Delta_{ij}^2/c^2)} \tag{2}$$

where $\Delta_{ij}$ is the distance between the $i^{th}$ and the $j^{th}$ spatial grid's cell centre and then the summation of the counts, $n_j$, over $j$ is done considering cells within distance equal to 3 times $c$ (being $c$ the aforementioned correlation distance) from the $i^{th}$ cell's centre.

Later, Woo (1996) proposed an alternative finite-range form for the kernel, based on the fractal dimension of epicentres and shown in Eq. 3:

$$\begin{cases} K(M, x) = \frac{D}{2 \cdot \pi \cdot h(M)} \left( \frac{h(M)}{r} \right)^{2-D} & \text{if } r \leq h(M) \\ K = 0 & \text{if } r > h(M) \end{cases} \tag{3}$$

where $M$ is magnitude for a location $x$, $r$ is the radial separation distance and $h(M)$ is a magnitude-dependent bandwidth parameter which can be parametrised as $h(M) = H \cdot \exp(k \cdot M)$ where $H$ and $k$ are regionally estimated constants using seismological and geological considerations and $D$ is the fractal dimension of the epicentres.

Subsequently, Helmstetter et al. (2006) proposed a model for the seismicity density calculation by means of an isotropic adaptive kernel (Izenman, 1991) that smoothed the seismicity depending on the number of events (in order to increase or decrease the detail in the seismicity calculations). Here, the parameter used for the smoothing kernel depends on the average distance between all the events around an earthquake but also, on the accuracy of the epicentre location in the first instrumental era of the earthquake catalogue.

Hiemer et al. (2014) created a model based on the seismicity and the fault moment release in order to consider the active mechanisms that generate seismicity in a more direct manner in order to smooth the seismicity (i.e., the locations of the earthquakes). They use the kernel defined by Helmstetter et al. (2006) and, similarly, the fault moment rate was smoothed with

an isotropic kernel (Eq. 4):

$$k_i(r) = \frac{C(d)}{(r^2 + d^2)^{1.5}} \cdot M_i \tag{4}$$

where $k$ is the value of the smoothing kernel for a fault point $i$, $M_i$ is the fault moment at that point, $d$ is the constant smoothing distance, $C(d)$ is the normalisation constant and $r$ is the epicentral distance.

More recently, the 2020 European Seismic Hazard model (ESHM20) has been released (Danciu et al., 2021). The authors combine the smoothing seismicity algorithms with active fault models. In this case, they point out the challenge of avoiding double counting events around faults when they consider the background seismicity and the one linked to the fault's activity. Another example of this approach is shown in the work by Pandolfi et al. (2023), where the authors combine 3D information of the seismic sources with the data in the seismic catalogue to calculate the seismic rate.

The works cited in the previous paragraph showcase the importance of considering the active seismogenic sources when computing the activity rate. A common assumption within PSHA is that seismicity can be well-described by a Poisson process (Cornell and Winterstein, 1987). A fundamental property of Poisson processes is that the instantaneous rate of events is constant and does not depend upon the occurrence of other events located close in either space or time. However, earthquake sequences feature a significant number of aftershocks, and these events are dependent upon the main shock. The purpose of declustering seismicity data is to remove these dependent events so that the underlying long-term average rate of occurrence can be estimated.

Taroni and Akinci (2021) proposed the use of aftershocks and foreshocks in the seismic activity calculation since removing such events from seismic catalogues may lead to underestimating seismicity rates and, consequently, the final seismic hazard in terms of ground shaking. To do this, they used as kernel a simple weight function of the form (Eq. 5):

$$k_i(N) = \frac{1}{N} \tag{5}$$

where $N$ is the number of events in the seismic series in which the event $i$ belongs.

This weight function ensures that the contribution of each event will be the same for the activity rate computation, regardless of its association with a seismic series.

With all the exposed factors, we are going to investigate the sensitivity of the activity rate computation model to both the proximity of the spatial cells to the seismic sources and the epicentral uncertainty related to each event of the catalogue and its influence on a time-dependent seismic hazard. Therefore, we will evaluate if the obtained values may be used as a decision factor on Operational Earthquake Forecasting (OEF). Additionally, we will consider the foreshocks and aftershocks in order to calculate this activity rate by means of a previous clustering process so each main shock and corresponding foreshocks and aftershocks are grouped in a given cluster, but we will also compare the results with the ones obtained by using a declustered catalogue. To do this, we will consider two case studies: Central Italy, a high seismicity area, which will help calibrate the models proposed, and South-eastern Spain, a moderate seismicity area, in which different treatments in the catalogue will be tested (declustering and using tectonic b-value vs time-dependent b-value).

## 2 Methodology

In this section, the procedure followed to obtain the parameters that will be used inside the smoothing kernel is described. The purpose of this kernel is to smooth the gridded seismic activity (for which a spatial grid is previously defined), helping to improve the description of the seismicity in the area. These parameters will define the different models to be tested in the different areas.

### 2.1 Smoothing kernel parameters

For this work, a modification of the kernel proposed by Frankel (1995) has been used to smooth the gridded seismicity (Eq. 6):

$$f(r) = A \cdot \exp\left(-\frac{(r-\mu)^2}{2 \cdot \sigma^2}\right) \tag{6}$$

where $r$ is the distance between the centre of the spatial grid cells and the centre of the cell in which the seismic activity is being computed, $A$ is the normalization constant, $\mu$ is the parameter that controls the $r$ value at which the maximum of the function is reached and $\sigma$ constraints dispersion of the function around the maximum value.

A more detailed review of the smoothing function, including some examples, can be seen in the section 2.1.3.

As expressed in the last paragraph of the introduction, we will avoid any arbitrary choice in the definition of these parameters ($\mu$ and $\sigma$) by assuming they have a geophysical meaning.

### 2.1.1 Geophysical meaning of the parameter $\mu$

The meaning of this parameter, within the context of the seismic activity smoothing, is the distance from a given cell centre to the point(s) in which the probability of having an earthquake is higher.

It is common to find that the value of this parameter is set to zero (Frankel, 1995; Helmstetter et al., 2006; Hiemer et al., 2014), as the maximum probability of having an earthquake is where it has already occurred before. So the smoothing function has its maximum value in the cell in which the seismic activity rate is being smoothed. This constitutes the first option regarding this parameter: $\mu = 0$.

An alternative model is proposed, where the maximum probability is set at the location of the nearest seismic sources. For this to be implemented, the minimum distance between the point in which seismic activity rate is being computed and the location of the nearest seismic source is calculated and named in this work from now on as $d_{f_i}$. So, the second option for the parameter value is $\mu = d_{f_i}$.

For areas in which the tectonic structures are only present in part of the region, a hybrid approach may be applied by using cut-off distance. This cut-off distance may be calculated as follows:

$$d_c = \overline{d} + 2 \cdot \sigma_d$$

where $d_c$ is the cut-off distance, $\overline{d}$ stands for the mean value of the distance between all the structures, and $\sigma_d$ is the standard deviation for all these distances.

If the distance from the centre of the spatial grid cell to the nearest fault is higher than the cut-off distance then $\mu = 0$. Otherwise, it will be set to $d_{f_i}$.

### 2.1.2 Geophysical meaning of the parameter $\sigma$

This parameter accounts for the dispersion of the values of the distribution around the mean value. That is to say, how far one might expect to find earthquakes around the most probable value (of distance). Therefore, we have considered that this second parameter is related to the accuracy of earthquake's epicentre measurement. This means that it would depend on the methodologies and instrumentation used for the calculation of the epicentre, and thus, on both the year and the location of the catalogue.

It should be noted that $\sigma$ may depend on other geophysical parameters such as the characteristics of ground, the style of faulting and/or the tectonic stress regime, to cite a few. Nevertheless, in this work only the influence of the uncertainty in the epicentre's location will be considered in the smoothing process.

As in the previous section, two different options regarding the epicentre uncertainty, $\varepsilon$, have been considered: either it depends on the year of occurrence ($\varepsilon_1$), or it is constant and computed as the mean value of the epicentral uncertainty for all the events ($\varepsilon_2$).

Three different models have been proposed to account for the variations in these parameters (Table 1), where $\varepsilon_1$, refers to the different epsilon values depending on the period of the catalogue, and $\varepsilon_2$ refers to the fixed value for all the catalogue:

**Table 1.** Models for the seismic activity smoothing.

| Parameters | Model 1 | Model 2 | Model 3 |
|:---:|:---:|:---:|:---:|
| $\mu$ | $d_{f_i}$ | 0 km | 0 km |
| $\sigma$ | $\varepsilon_1$ | $\varepsilon_1$ | $\varepsilon_2$ |

### 2.1.3 Examples of the smoothing kernel implementation

In this section, some examples of how the smoothing kernel works are shown. Here, the seismic sources are faults in a shallow seismicity context, so the trace of such faults has been considered as the location with the maximum probability of having an earthquake. This approach has also been considered for the two case studies in this work. Three main scenarios have considered to showcase the smoothing kernel:

■ Usual 1D Gaussian filter, $\mu = 0$

This is the case when using models 2 and 3 also when the distance from the centre of the spatial grid cell in which the seismic activity rate is computed to the nearest fault is greater than $d_c$ as defined in the section 2.1.1. An example can be seen in Figure 1a.

■ Single fault, $\mu \neq 0$

When the nearest fault is closer than $d_c$ from the centre of the spatial grid cell then the resulting function will provide a ring-shaped smoothed activity, the width of which will depend on $\sigma$. Only the section of this ring in which the fault is located will be used in the smoothing. This can be achieved by considering the $n$ closest points to the spatial grid cell centre and then computing the angles to define the ring arc (Figure 1b).

■ Several faults, $\mu \neq 0$

This case is a generalization of the former with the exception that when spatial grid cell's centre is in between faults and at similar distances, then the full ring will be used as smoothing function Figure 1c. On the other hand, if the distance to both faults is similar but the spatial grid cell's centre is not in between the faults then the resulting smoothing is a ring arc (Figure 1d).

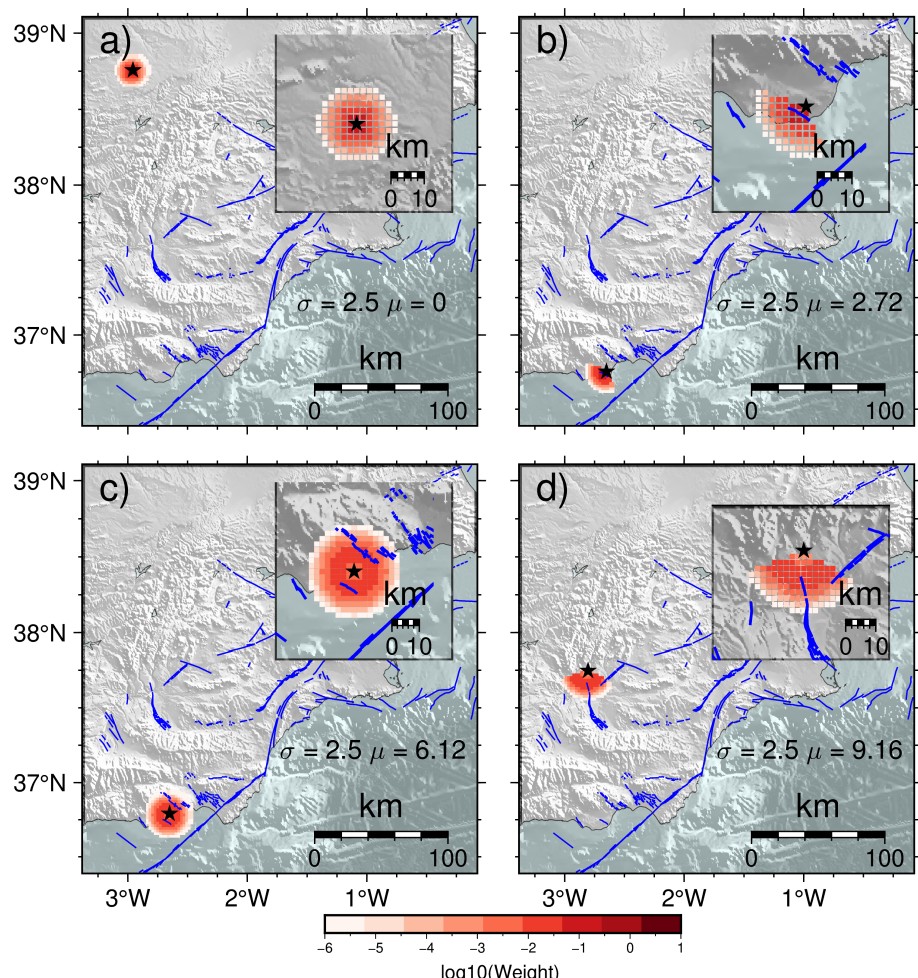

**Figure 1.** a) Smoothing function for $\mu = 0$. This can happen when either the distance is greater than $d_c$ or when the spatial cell is over the seismic source. b) Smoothing function for $\mu \neq 0$ and a single fault, i.e. when only one seismic source is present and within distance lesser than $d_c$. c) Smoothing function for $\mu \neq 0$ when several seismic sources are surrounding the spatial grid cell at similar distances (with angular amplitude greater than or equal to 180º) . d) Smoothing function for $\mu \neq 0$ in the case that several seismic sources are surrounding the spatial grid cell at similar distances (with angular amplitude lesser than 180º). The blue lines show the fault traces. In this example $d_c$ equals 48 km. The stars mark the spatial grid cell considered in each case.

## 2.2 Cluster identification and seismicity smoothing

First, the spatial grid is defined by creating a rectangle spanning the maximum and minimum longitudes and latitudes of the catalogue with the desired resolution. The choice of the resolution can be motivated by similar studies in comparable tectonic settings or the order of the epicentral uncertainty of the earthquakes in the catalogue. For this work, in the case of Spain the same resolution as in a previous work in the same area by Montiel-López et al. (2023) has been used. In the case of Italy,

although Murru et al. (2016) use a 0.025ºx0.025º grid and Gulia and Wiemer (2019) use a 2-km spaced grid, we decided to
use the same resolution for both case studies, a 0.015ºx0.015º grid. All the data used in this work can be found in the "Data availability" section.

Then, all the events of the catalogue must be assigned to each cell. This is done by calculating the minimum distance of each event to all the spatial grid cell's centres.

One of the most important steps regarding the activity rate calculation in this work, is the identification of the seismic clusters present in the area for the selected period of time. As indicated in the introduction, we do not pretend to remove the foreshocks and aftershocks but to identify the main event and all related events in the corresponding cluster.

To do so, even though Epidemic Type Aftershock Sequence (ETAS) model allows to assign to each event the probability of being an aftershock (Zhuang et al., 2002; Marsan and Lengliné, 2008; Console et al., 2010) in this work we have decided to select a non-stochastic method based on the performance classifying events of a relevant seismic series.

There are several options for this task: a) using the Reasenberg and Jones (1989) algorithm -RJ by applying the ZMAP software (Wiemer, 2001), or b) or using the Afteran -A- algorithm (Musson, 1999) or the Gardner and Knopoff (1974) -GK74- declustering algorithms by applying the Python libraries included in OpenQuake's python scripts (Pagani et al., 2014).

These algorithms will flag each event from each cluster with an identifier which will be added as a column to the catalogue, and the events that do not belong to any series will have a value of zero for this field. In order to decide which algorithm performs best on the data, a comparison between the RJ, A, and GK74 declustering algorithms has been made using default parameters (see section 3.2.1). An example can be seen in Figure 2.The smoothing procedure explained by Taroni and Akinci (2021) has been adapted to work with as many clusters as can be found in each spatial grid cell.

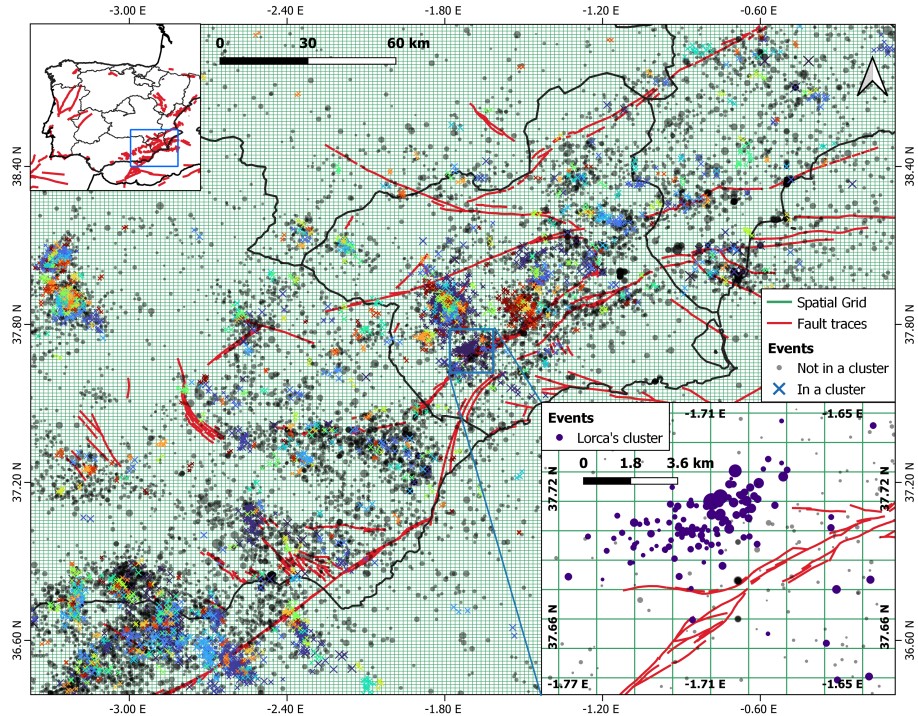

**Figure 2.** Example of earthquake clusters identified in South-eastern Spain. All the cross markers with the same colour indicate events that belong to the same cluster, while the grey circles are events not belonging to any cluster. The fault traces are presented in red-coloured lines and the spatial grid cell's limits are green lines. A zoom in on the Lorca's cluster is shown in the bottom right corner, where the events are plotted using purple circles, the size of which depends on the magnitude of the event.

To do this, all the events belonging to each cluster are counted and define the cluster weight, $c_j$. If an event does not belong to any cluster (i.e., the cluster label for that event is set to zero) then $c_j$ equals 1. The weighted counts for each spatial grid cell are calculated as the summation of all the events over the different clusters (Eq. 7):

$$k_i = \sum_1^j \left( \frac{1}{c_j} \sum_1^m 1 \right) \tag{7}$$

where $k_i$ is the weighted count of events inside the cell $i$, the first sum goes over the number of clusters, $j$, inside the $i-th$ cell, and the second summation goes over each event, $m$, of the cluster $j$ that is inside the spatial grid cell $i$.

For instance, if inside a cell there are 20 events that belong to a cluster composed of 100 events in total and 13 events that do not belong to any cluster, the weighted number of events for that cell will be $k_i = 13 + 20/100 = 13.2$.

## 2.3 Seismic activity rate computation

After the smoothing process explained before, we will consider both the nature and source of the earthquakes and the uncertainty related to the earthquake location.

Thus, the seismic activity rate is calculated as the product of the weighted counts and the smoothing kernel values (Eq. 8):

$$\lambda_i = \mathbf{w_i} \cdot \mathbf{k} \tag{8}$$

where $w$ is an $n \times n$ matrix, being $n$ the number of cells of the spatial grid, that for each cell, $i$, contains the $n$ values of the smoothing kernel associated to the cell. On the other hand, $\mathbf{k}$ is a vector containing the weighted count for each cell $i$ as defined in Eq. 7. So, for each cell the vector product between the smoothing kernel ($\mathbf{w_i}$ can be seen as a vector) and the weighted count is done. This means all cell counts are added in each cell activity rate computation and the smoothing function works similarly to the correlation distance proposed by Frankel (1995).

## 2.4  Exceedance probability computation

The annual exceedance probability of a given PGA has been obtained by developing a Python script based on OpenQuake (Pagani et al., 2014). Two models have been tested for the computation of the needed b-value: a fixed (time-independent) b-value assigned from the tectonic zones of each country and a gridded (time-dependent) b-value calculated using the methodology proposed by Montiel-López et al. (2023). In both cases, another Python script was developed to obtain a gridded time-dependent seismic activity rate for moment magnitudes greater than or equal to 4.0 Mw for each cell.

The Akkar and Bommer (2010) empirical equation has been used as a ground motion prediction model since it is appropriate for Mediterranean regions as Spain, whereas the Akkar et al. (2014) ground motion empirical equation has been used for Central Italy since it was used in the EHSM20 (Danciu et al., 2021). An example of the workflow for this work is represented in Figure 3.

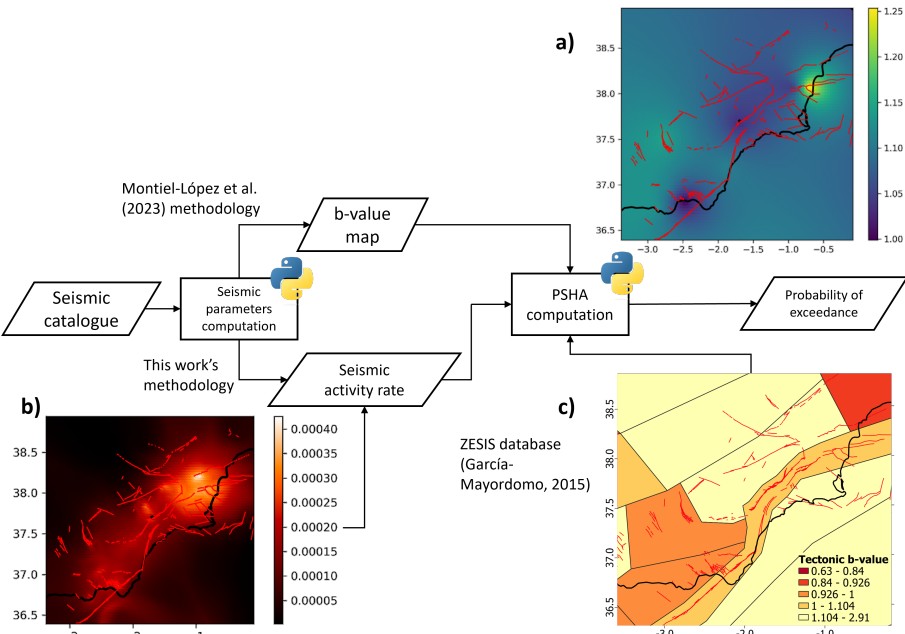

**Figure 3.** Workflow diagram for the exceedance probability computation. Examples of the main inputs are given as the three spatial mappings of a) b-value, b) seismic activity rate and c) tectonic zones' b-value.

Our goal is to investigate if the changes in the seismic activity and b-value time series can be observed as a trend in the PSHA results. In the case such trends are observed, this methodology could be used for OEF. For this reason, the temporal evolution of the annual probability of exceedance (PoE) of a background PGA corresponding to a 475 years return period (i.e., 0.002 PoE) has been computed as a time-dependent value. The results have been expressed as a relative change (RC in percentage change, Eq. 9) between background annual exceedance probability (long-term value) and the time-dependent annual exceedance probability. Depending on the country, the background (long-term) PGA value may have been updated in the corresponding seismic hazard studies. This could be due to the occurrence of new damaging earthquakes or improvements in the seismic knowledge of a region, amongst other reasons. In case such changes have been made, a new background PGA value has been computed using the data up until the year the seismic hazard information was updated.

$$RC = 100 \cdot \left( \frac{PoE}{0.002} - 1.0 \right) \tag{9}$$

In order to save computation time, the annual exceedance probability is only calculated for a selection of cities located inside the spatial grid.

Additionally, we have computed the annual variation of the RC in the exceedance probability ($RC_i - RC_{i-12}$, with $i$ the computed month) and the monthly variation ($RC_i - RC_{i-1}$) to investigate if any of these metrics is effective as an indicator for OEF.

## 3 Case studies

As explained before, the goal of our smoothing methodology is to test the viability of producing time-dependent seismic hazard results which may be used for taking decisions before the main earthquake. Therefore, now we will present and discuss the results obtained for two different regions with different seismic behaviour, Central Italy (high seismicity) and south-east Spain (low seismicity). We will check if there are significant changes in the defined metrics before the occurrence of important earthquakes carrying out a retrospective validation of how useful the results are.

### 3.1 Central Italy

#### 3.1.1 Catalogue preparation and parameters for computation

As mentioned before, since Central Italy is a very active region, this case study will help us to decide which of the models (Table 1) performs better. Central Italy (Abruzzo, Campania, Lazio, Marche, Moise, Toscana and Umbria) is a region where several high magnitude earthquakes and significant seismic series have occurred in the past. The main focus is on L'Aquila, where a 6.29 Mw earthquake (Table 2) struck the area in 2009 and caused 309 deaths and 1500 injured. Therefore, the city of L'Aquila has been selected as the site for the hazard computation.

**Table 2.** L'Aquila earthquake data and distance to hazard computation site.

| Location | Lat. (ºN) | Long. (ºE) | Depth (km) | Mw | Int. (EMS-98) | Date | Epicentral distance (km) to L'Aquila |
|---|---|---|---|---|---|---|---|
| L'Aquila (AB) | 42.342 | 13.380 | 8.3 | 6.29 | VIII | 6 Apr 2009 | 5.2 |

Figure 4 shows the location of the area of study (a rectangle with longitudes from 11.394 to 15.391º E and latitudes from 40.359 to 44.353º N) and the tectonic zones and main faults as defined by Danciu et al. (2021) to compute the European Seismic Hazard Map. For this area, the Italian HORUS (Lolli et al., 2020) catalogue has been used as it has been homogenised and comprises events from 1960 to 2023. This catalogue is a homogenised instrumental catalog based on the hypocentral location of earthquakes compiled from the Italian Seismological Instrumental and parametric Database (ISIDe) (ISIDe Working Group, 2007). A spatial filtering process has been applied to the catalogue to extract the events within the above-mentioned area, so 49112 events with maximum depth of 30 km and maximum magnitude of 6.81 Mw remain.

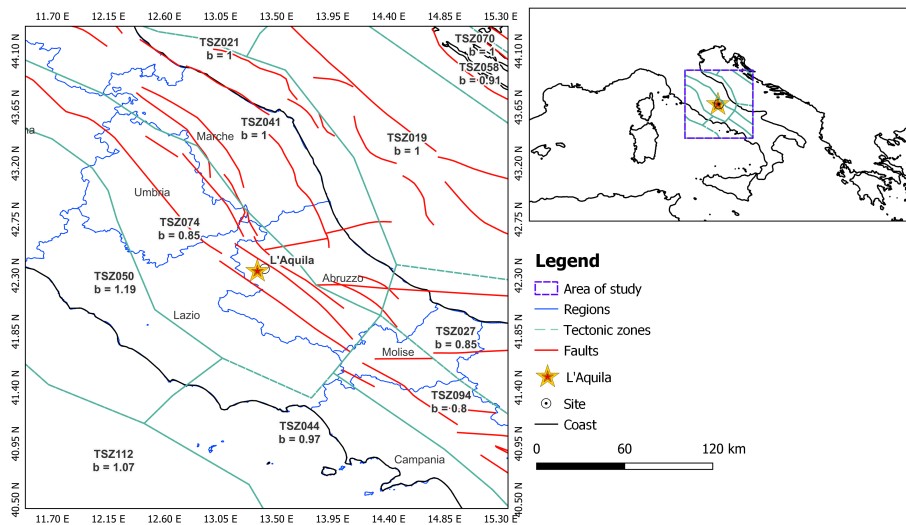

**Figure 4.** The map shows the tectonic zones (green lines) in Central Italy with their acronyms and tectonic b-values as can be found in Danciu et al. (2021). The star marks the epicentre of L'Aquila earthquake (Table 2) and the red lines represent the fault traces.

The events that are not earthquakes (such as quarry blasts, eruptions, explosions, etc.) have been filtered out from 2012 on (as the catalogue has such information). In order to consider the influence of such events prior 2012 the area that has been selected for this study does not show important changes in the b-value according to the results of Taroni et al. (2022). In this case the catalogue has not been declustered, but the clusters have been identified by using the GK74 algorithm and this information has been used to weight down the influence of the non-independent events towards the seismic parameters' computation.

A spatial cell grid of 0.015º×0.015º spanning the above longitude and latitude ranges has been created (using 70756 points). The completeness magnitude (Table 3) has been retrieved from Taroni et al. (2021).

**Table 3.** Completeness magnitude values proposed by Taroni et al. (2021).

| Year | 1960 | 1980 | 1990 | 2003 | 2005 |
|---|---|---|---|---|---|
| **Completeness magnitude (Mw)** | 4.0 | 3.0 | 2.5 | 2.1 | 1.8 |

Since the $\sigma$ parameter used in the smoothing kernel computations and based on the location uncertainty aims to account for the physical variability in the location of the earthquakes, three models with different uncertainty values have been tested to showcase the variability in the results as a consequence of increasing or decreasing the uncertainty of the epicentre location. In order to obtain such values, the work from (Scudero et al., 2021) gives insight on the variation of the horizontal error (ERH) in Italy as well as a range of mean values for different revision processes on the data (2.2, 3.3 and 13.1 km). Given that the HORUS (Lolli et al., 2020) catalogue, used in this work, has no information on the ERH, but the locations of the events are obtained

through the ISIDe database, their spatial uncertainty can be deduced from the CPTI15 catalogue (Rovida et al., 2020, 2022).
The aforementioned range of mean values for the ERH is coherent with the mean spatial uncertainty obtained from the CPTI15 catalogue.

Therefore, a minimum value of 6 km, in agreement with the previous explanation, and a maximum value of 30 km, following the work of Taroni et al. (2021) has been chosen to characterise the spatial uncertainty. The three models proposed for the seismic activity smoothing are presented in Table 4.

**Table 4.** Models for the activity smoothing in Central Italy.

| | Time-dependent models | | |
| --- | --- | --- | --- |
| **Parameters** | **Model 1t** | **Model 2t** | **Model 3t** |
| $\mu$ | $d_{f_i}$ | 0 km | 0 km |
| $\sigma$ | 6 km | 6 km | 30 km* |

*From Taroni et al. (2021).

### 3.1.2   Results

Figure 5 (Model 1t) presents a moderate increase in the annual exceedance probability (25%) one month before L'Aquila earthquake occurred, and not only the annual but also the monthly variations of relative change reaches values higher than 35%. This sudden change is due to the foreshock activity that preceded the main-shock, as a 4.1 $M_L$ earthquake occurred on 30 March 2009. Figure 6 (Model 2t) shows a similar trend in all the metrics as the previous model with a slightly lower value
for the exceedance probability change before the earthquake (22%) and the annual and monthly variations (32%). The Model 3t (Figure 7) provides the lowest values for the metrics (-3%, 4% and 3%, respectively). After this increase the RC slowly decreases over time for all three models.

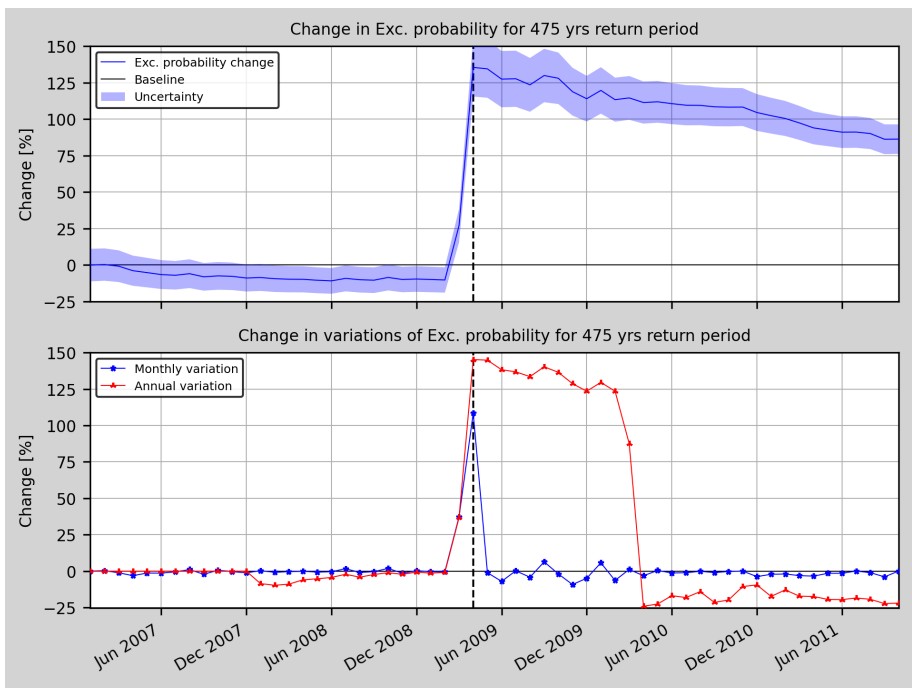

**Figure 5.** Model 1t. Relative change (RC) of the annual exceedance probability (top) and its annual and monthly variation (bottom). The vertical dashed black line marks the occurrence of L'Aquila earthquake.

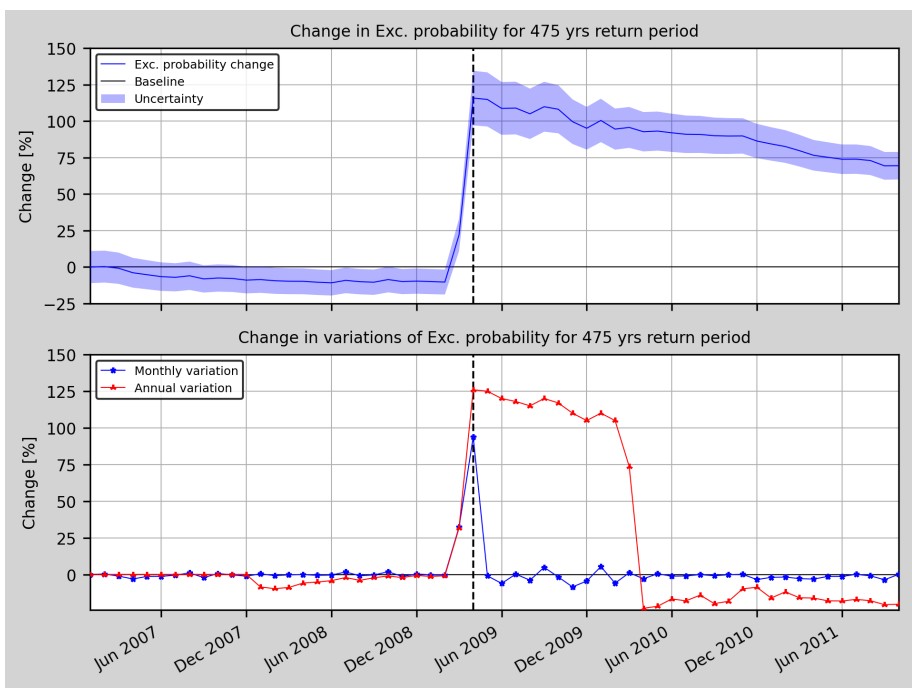

**Figure 6.** Model 2t. Relative change (RC) of the annual exceedance probability (top) and its annual and monthly variation (bottom). The vertical dashed black line marks the occurrence of L'Aquila earthquake.

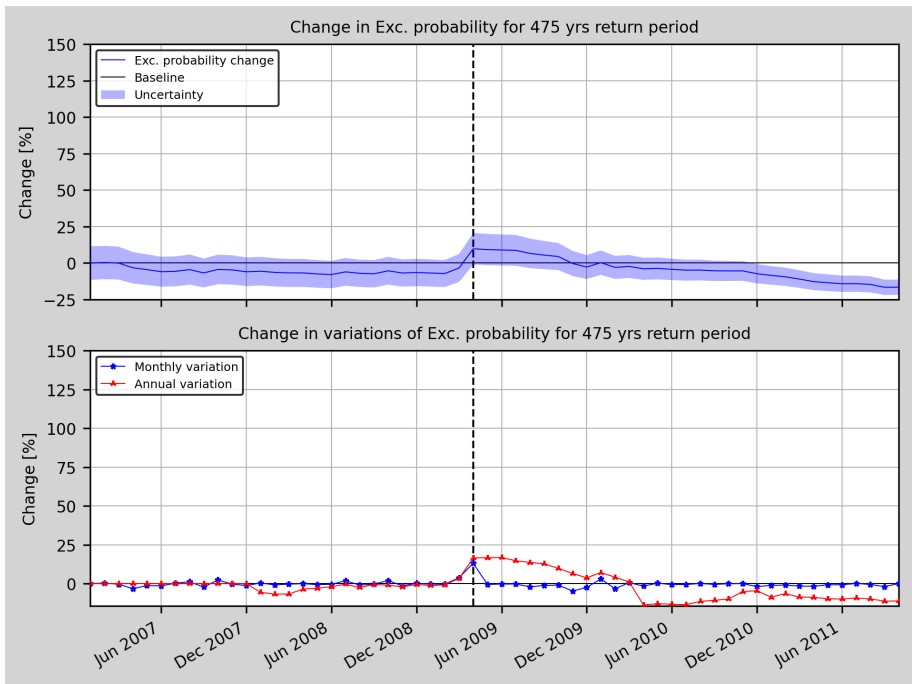

**Figure 7.** Model 3t. Relative change (RC) of the annual exceedance probability (top) and its annual and monthly variation (bottom). The vertical dashed black line marks the occurrence of L'Aquila earthquake.

Given that the main objective is to be able to perform OEF in the area of study, the Model 1t (Figure 5) has been selected since it performs better in terms of exceedance probability change. Moreover, its annual and monthly variations are the highest one month before the main shock (when compared with the models 2t and 3t).

## 3.2 South-eastern Spain

### 3.2.1 Catalogue preparation and parameters for computation

The south and south-east of Spain are the regions with a higher seismic hazard in Spain (IGN-UPM Working Group, 2013; Kharazian et al., 2021) with values reaching 0.23 g for a 10% probability of exceedance in 50 years (i.e., return period of 475 years). Although Spain is a moderate to low seismic region compared to other European countries such as Italy or Greece, it has been exposed to several damaging earthquakes in the past being the most representative the 1829 *Torrevieja* earthquake and the 1884 *Arenas del Rey* earthquake, both with a maximum intensity IX-X. Additionally, in the last 25 years, South-eastern Spain has suffered seven earthquakes with Mw greater than or equal to 4.5 (Table 5 and Figure 8 present only those classified as main-shocks and located in the area of study), being the 2011 *Lorca* earthquake the most relevant since it was the most recent earthquake causing damage to buildings and injuries to the population. The seismicity is usually very shallow (mainly lower than 10 km). Three main cities (Murcia, Lorca and Vera from North to South) have been chosen as representative of the region in terms of decreasing seismic hazard values for a 475 years return period.

**Table 5.** Damaging earthquakes in the last 25 years and epicentral distance to some chosen cities in the area of study.

| Location | Lat. (ºN) | Long. (ºE) | Depth (km) | Mw | Int. (EMS-98) | Date | Epicentral distance (km) to | | |
|---|---|---|---|---|---|---|---|---|---|
| | | | | | | | Murcia | Lorca | Vera |
| N. Mula (MU) | 38.0963 | -1.5014 | 1.1 | 4.86 | VI | 2 Feb 1999 | 34.5 | 49.9 | 101.4 |
| S. Gergal (AL) | 37.0931 | -2.5379 | 0.8 | 4.60 | V | 4 Feb 2002 | 159.4 | 48.5 | 62.0 |
| SW. Bullas (MU) | 37.8925 | -1.8353 | 1.2 | 5.00 | V | 6 Aug 2002 | 62.9 | 27.1 | 73.7 |
| NW. Aledo (MU) | 37.8535 | -1.7555 | 10.9 | 4.80 | VII | 29 Jan 2005 | 20.6 | 57.0 | 69.9 |
| Lorca (MU) | 37.7175 | -1.7114 | 4.0 | 5.10 | VII | 11 May 2011 | 59.5 | 5.1 | 55.7 |

In order to compute the seismic activity rate to be used in a PSHA, first we need to compile a homogeneous and complete seismic catalogue in the influence area, needed for the chosen locations. This catalogue comprises, obtained from the *Insituto Nacional de Geografía* webpage (Instituto Geográfico Nacional (IGN)) all the events from 1396 to August 2023 in South-eastern Spain inside the tectonic zones of *Eastern Betic Shear Zone (ZCBOR), Eastern Inner Betics (BIOR), Valencian Plateau and Alicante's Prebetic (PVPA), Murcian Prebetic (PM), Sierra Nevada-Filábrides and Guadix-Baza (SNFCGB), Central Inner Betics (BIC), Southern Plateau (MS), Cazorla-Segura and Albacete's Prebetic (CSPA), Central Guadalquivir and Algerian-Balearic Basin (CAB)*, as defined by García-Mayordomo (2015) to create the Spanish Seismic Hazard Map (IGN-UPM Working Group, 2013).

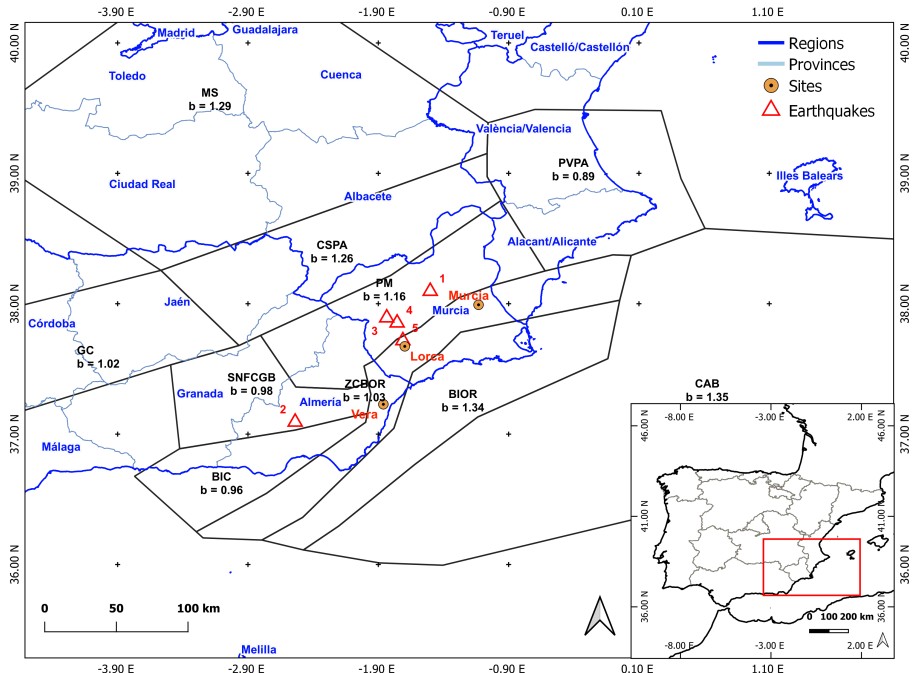

**Figure 8.** Map showing the tectonic zones in the South-eastern Spain region with their corresponding b-values (black numbers) as computed by García-Mayordomo (2015). The red triangles mark the earthquakes and the red numbers the order in which they appear in Table 4. The orange circles locate the main sites for the seismic hazard analysis.

The catalogue contains a total of 20279 events that span from 1396 to August 2023. Their moment magnitudes range from 0.1 to 6.8 after being homogenised using the magnitude correlation equations for this region (IGN-UPM Working Group, 2013). Their depth goes up to 90 km, although in the calculations only the earthquakes shallower than 30 km are considered (which amount for a total of 20168 events). A spatial grid of 0.015°x0.015° covers the area of study (using 40401 points).

Table 6 presents the number of clusters and the events in clusters for the whole seismic catalogue. The RJ algorithm classifies a total of 652 clusters in the catalogue while GK74 detects 1012 clusters. The A algorithm identifies 1245 clusters. As can be seen, despite the three methods relying on windows for their calculations, there are significant differences in the results, not only in the number of clusters but also in the number of events inside each cluster.

Zaliapin and Ben-Zion (2020) pointed out these problems with the identification of aftershocks and main shocks and proposed an algorithm to discriminate between background and clustered events by randomly thinning a complete catalogue by removing nearest-neighbour earthquakes. Moreover, Anderson and Zaliapin (2023) examine the effect on the hazard estimation when using different declustering thresholds. They conclude that hazard estimates are most sensitive to the catalogue thinning near the aftershock zone, and less sensitive elsewhere.

**Table 6.** Comparison of cluster identification and total events inside clusters among three declustering algorithms: an analysis using Lorca's seismic series.

| Algorithm | RJ | A | GK74 |
|---|---|---|---|
| **Number of clusters** | 652 | 1245 | 1012 |
| **Events in clusters** | 7143 | 10167 | 7552 |
| **Events inside Lorca's series** | 123 | 196 | 136 |

In spite of the difficulties in defining the clusters, Cabañas et al. (2011) carried out a detailed study on the 2011 Lorca's earthquake seismic series. This study is the best definition at the moment so we will use it to validate the best algorithm. They identified 146 events (including the foreshock, the main shock and the aftershocks) that belong to Lorca's series, from 11 May 2011 until 19 July 2011. With this information, in order to test the performance of the declustering methods, the confusion matrices for each one have been computed. In the area of study, a total of 249 events have been recorded, which means a total of 103 background events should be identified. For this analysis, all the events classified in a cluster different from the one of Lorca series have been considered as background for simplicity. Figure 9 shows that GK74 method is the most adequate (with a 94.43% mean for the metrics compared with the 92.88% for RJ and a 78.79% for A) and also the one that is able to identify more events belonging to Lorca's series.

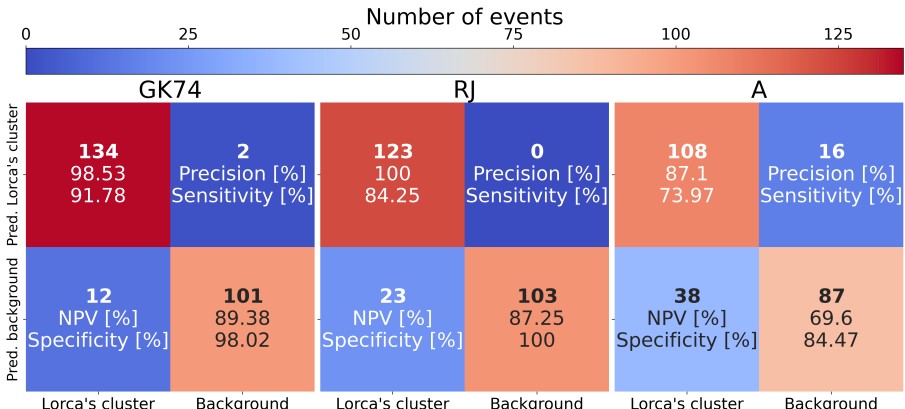

**Figure 9.** Confusion matrices for the tested declustering methods. Inside each square, the number of events (bold) and some metrics computed using the data are presented (NPV stands for Negative Predictive Value).

The catalogue starts in the historical period (when there was a lack of instrumentation and procedures to accurately locate the epicentres and evaluate the magnitude of the earthquakes) and ends in the present days. This implies that not all the magnitude values will be complete in the catalogue (low magnitudes are missing in the historical period) and the location uncertainty will

also differ depending on the year of detection. First, we will characterise the completeness magnitude -the minimum magnitude from which the catalogue is not missing any record- and periods for the Spanish seismic catalogue.

Gaspar-Escribano et al. (2015) defined different threshold magnitudes for different regions around Spain. The class marks of these magnitude intervals for the zone of interest (South-eastern Spain) have been selected as the completeness magnitudes up until 1962. From 1962 on, the completeness magnitude has been computed by spatially averaging the gridded completeness magnitude results available from González (2017) over the area of study. The values used in this work are presented in Table 7.

**Table 7.** Completeness magnitude for each period according to Gaspar-Escribano et al. (2015) in the top tabular and the spatially averaged completeness magnitude for each period using the results from González (2017) in the bottom tabular.

| Completeness magnitude (Mw) | 6.25 | 5.75 | 5.25 | 4.75 | 4.25 |
|---|---|---|---|---|---|
| From year | 1048 | 1521 | 1801 | 1884 | 1909 |
| to year | 1520 | 1800 | 1883 | 1908 | 1962 |

| Completeness magnitude (Mw) | 3.40 | 3.30 | 3.00 | 2.90 | 2.30 | 2.10 | 1.90 | 1.80 |
|---|---|---|---|---|---|---|---|---|
| From year | 1963 | 1980 | 1985 | 1993 | 1999 | 2003 | 2011 | 2014 |
| to year | 1979 | 1984 | 1992 | 1998 | 2002 | 2010 | 2013 | 2023 |

The uncertainty of the epicentral location ($\varepsilon$) varies with time, showing a decreasing behaviour since the techniques and instrumentation have continuously been improved. The appropriate estimation of this uncertainty is very important in order to correctly assign the location of each earthquake to a given seismic source.

Following the research of Peláez and López Casado (2002), the $\varepsilon$ values for each period are presented (Table 8). The value corresponding to the period 1990-2023 has been obtained as the average epicentral uncertainty using the data provided by the national seismic network. A second fixed $\varepsilon$ value of 7.5 km has been computed as the mean value for all the uncertainties in the catalogue.

**Table 8.** $\varepsilon$ values proposed by Peláez and López Casado (2002).

| Period (yrs) | 1396 - 1700 | 1700 - 1920 | 1920 - 1960 | 1960 - 1990 | 1990 - 2023* |
|---|---|---|---|---|---|
| $\varepsilon$ (km) | 20 | 15 | 10 | 5 | 2.5* |

\* Calculated as the average epicentral uncertainty for the 1990-2023 period events in our catalogue.

The three models to be evaluated are presented in Table 9. Fixed model implies a fixed b-value (the b-value from the tectonic zone) while time-dependent model indicates a time-dependent b-value.

**Table 9.** Models for the exceedance probability calculation in South-eastern Spain.

| Parameters | Fixed models | | | Time-dependent models | | |
|---|---|---|---|---|---|---|
| | Model 1f | Model 2f | Model 3f | Model 1t | Model 2t | Model 3t |
| $\mu$ | $d_{f_i}$ | 0 km | 0 km | $d_{f_i}$ | 0 km | 0 km |
| $\sigma$ | Table 8 | Table 8 | 7.5 km | Table 8 | Table 8 | 7.5 km |

### 3.2.2 Results

After computing the time-dependent PSHA for the different models shown in Table 9, we have observed that although all the graphs show similar behaviour for the RC, Model 1t provides greater annual and monthly variations of the RC for some of the earthquakes than the rest, similarly to Italy's case study. Therefore, with the exception of the fixed models (models 1f, 2f and 3f) where we present a general comparison between them, we will present the results of Model 1t in this section along with figures comparing all three models. The stand-alone figures for the models 2t and 3t can be found in the Appendix section.

□ *Time-dependent PSHA using tectonic zones' b-values (fixed model)*

The time-dependent PSHA (PGA for a return period of 475 years) has been computed using the proposed methodology, for the compiled non-declustered catalogue, in one-month increments starting from 1990. The b-value is constant and given by the zonation proposed by García-Mayordomo (2015) (models 1f, 2f and 3f). This background PGA value will be a long term PSHA which varies each time that the seismic normative has changed in Spain. Our first background PGA corresponds to the PSHA computed using a catalogue with the same length as the one used for the NCSE-94(1994). This background PSHA value will be used from 1990 to December 1998 (since the next code updated the seismic hazard map using a seismic catalogue up to 1999). The second background value will correspond to the PSHA computed with a catalogue of the same length as the one used for the NCSE-02 (2002) and it will be used from 1999 to May 2011 (since that is the year when the seismic hazard map was updated again). Finally, the last background value, corresponds to PSHA computed using a catalogue of the same length as the one used for the current seismic hazard map for Spain (IGN-UPM Working Group, 2013). This last value is used from June 2011 to August 2023.

Figure 10 represents the temporal evolution of the results for each tested model. The vertical lines correspond to the main earthquakes from Table 5 and with epicentral distance lower than 75 km from the chosen city, since more distant events would not require a forecasting as they are not expected to cause damage. As can be seen, the behaviour is similar for the three models. The exceedance probability decreases continuously since 1990 except for Lorca in 1997, and 2006; Murcia in 1996; and Vera in 1994 and 1999. These variations are due to seismic activity changes, but they do not appear to be related with the occurrence of any of the main earthquakes from Table 5. On the other hand, all the models provide similar changes in the exceedance probability although in the city of Lorca, Model 3f is the one with the lowest percentage change and Model 2f is the one with the highest percentage change.

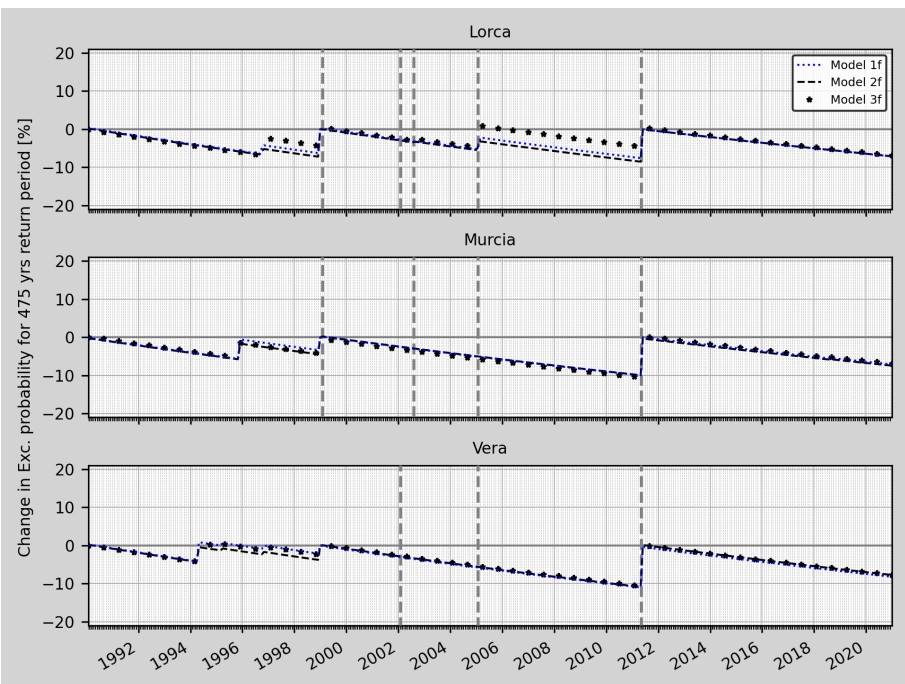

**Figure 10.** Relative change (RC) of the annual exceedance probability for models 1f, 2f and 3f. The vertical dashed grey lines mark the earthquakes considered in Table 5 which are closer than 75 km to each one of the sites.

It seems that the use of a constant b-value coupled with a time-dependent seismic activity rate leads to a RC that decreases over time with a constant rate and sudden increases mainly due to changes in the background PGA values. This behaviour is
380 due to the update of this parameter depending on the period of the catalogue, which is increased a month at a time (as it was the selected minimum time step for this area). We find that this approach is not appropriate towards earthquake forecasting for areas with low to moderate seismicity. This uniform behaviour potentially rules out the possibility of finding any metrics for OEF.

□ *Time-dependent PSHA using time-dependent b-value (time-dependent model)*

In this section, models 1t, 2t and 3t Table 9 are tested using the three PGA background values explained previously (and computed in January 1990, December 1998 and May 2011). As it can be seen in Figure 11, the annual probabilities decrease before Mula earthquake for Lorca site. However, close to the occurrence of the earthquake, it shows a slight increase even in Vera site, although it is 101.4 km away from the earthquake's epicentre. In Murcia site, the RC continuously decreases until five months before the earthquake, when it shows a sharp increase from -75% to -60% in the change of exceedance probability.
This change is also seen in the annual and monthly variations of the RC (Figure 11 zoom in). After Mula earthquake the change in probability exceedance remains higher than 20% (even increasing until 50% in the case of Lorca and 100% in the case of Murcia) for both Lorca and Murcia sites until Lorca earthquake happens. In Vera site, this parameter oscillates about the baseline. After 2011, the RC steadily increases in Vera, whereas in Lorca and Murcia it stays constant after 2019. On the other

hand, Model 2t and Model 3t (Figure 12) show a similar behaviour to Model 1t. It should be noted that the sudden changes in the RC in January 1999 and one month after the Lorca earthquake, i. e., the -60% to 0% increase in January 1999 (for both Lorca and Murcia) and the 100% to 0% decrease in June 2011 (for both Lorca and Murcia), are artefacts due to the change in the background PGA and cannot be considered in the analysis. However, the annual and monthly variations of the RC shown in Figure 13 and Figure 14 allow to see the changes related to the aforementioned earthquakes occurrences.

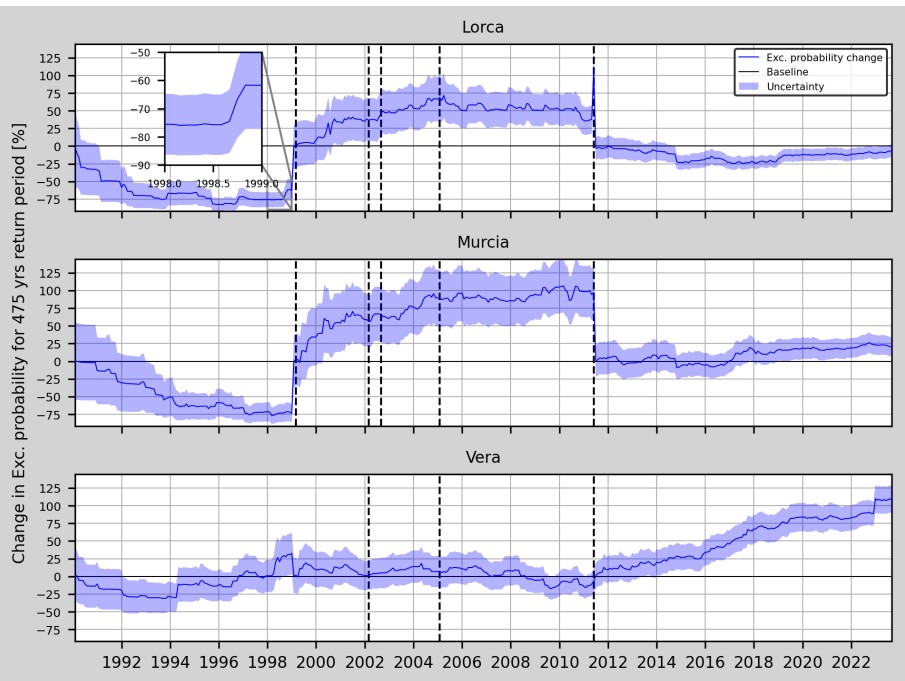

**Figure 11.** Relative change (RC) of the annual exceedance probability and corresponding uncertainty for Model 1t in Lorca, Murcia and Vera (from top to bottom). The vertical dashed black lines mark the earthquakes considered in Table 5 which are closer than 75 km to each one of the sites. A zoom in on the mentioned increase in the RC during 1998 in Lorca's site appears in the upper left side of the graph.

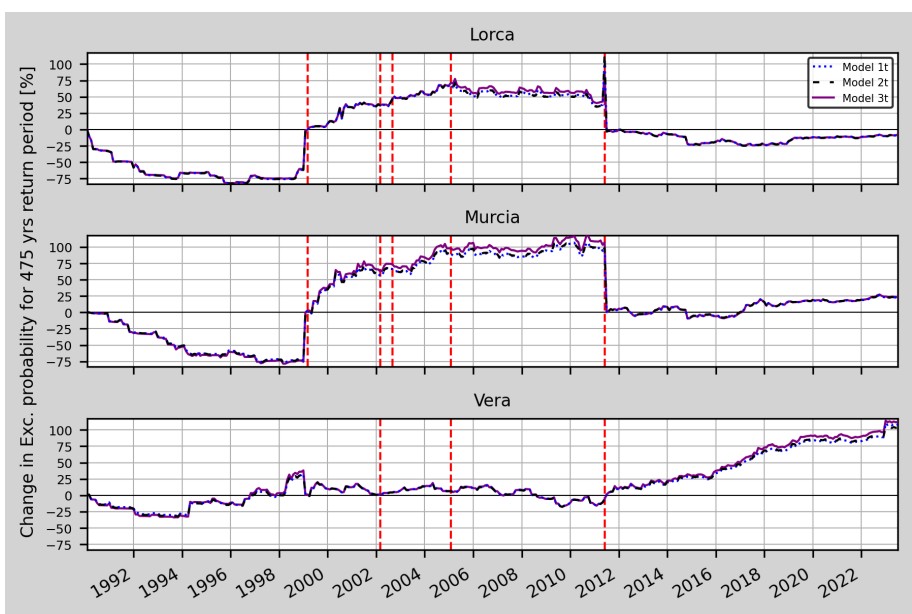

**Figure 12.** Mean value of the relative change (RC) of the annual exceedance probability for models 1t, 2t and 3t in Lorca, Murcia and Vera (from top to bottom). The vertical dashed red lines mark the earthquakes considered in Table 5 which are closer than 75 km to each one of the sites.

Figure 13 shows a 20% mean decrease in the annual variation of the RC from January 1991 to March 1993 that could be explained by the RC uncertainty (as can be seen in Figure 11 for both Lorca and Murcia sites, with higher uncertainty for that period). Then, a 15% increase in the annual variation can be seen in the RC from October 1998 until August 1999 for Lorca site (three months before Mula earthquake and then six months after it). This increase can also be seen in the declustered catalogue scenarios (Figure 16, Figure A5 and Figure A6). In Vera site, the increased RC variation during 1999 could be due to changes in the b-value from April to December (6 earthquakes with magnitude from 3.5 to 3.8 Mw occurred). In the case of Gergal and Bullas earthquakes in 2002, an increase in the variation of the RC cannot be observed. However, it can be seen for Lorca and Murcia sites that from July 1999 to May 2001 that the annual variations of the RC reach values higher than 15% with respect the baseline and with a mean value of 10% over this period. This increased values cannot be related with any close seismic activity greater than or equal to 4.0 Mw. It can also be seen that in Lorca site the annual variation stays higher than 20% for one year after Lorca earthquake. Lastly, the peak in the annual and monthly variation at Vera in 2022 appears due to the seismic activity in Turre (a town 14 km south from Vera) where a 4.0 Mw earthquake struck on 31 December 2022.

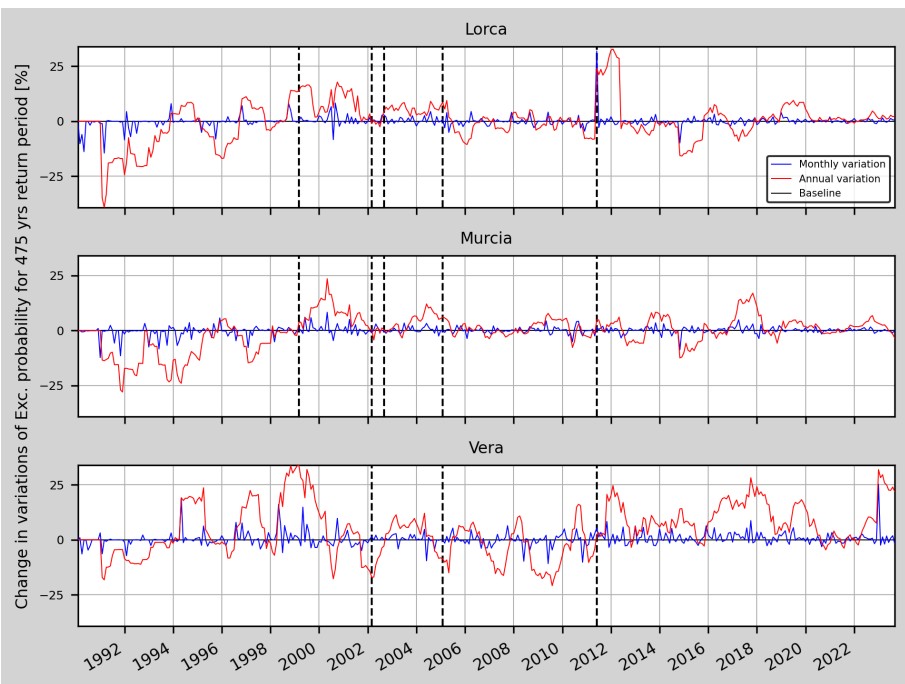

**Figure 13.** Annual and monthly variations of the relative change of the annual probability of exceedance for Model 1t in Lorca, Murcia and Vera (from top to bottom). The vertical dashed black lines mark the earthquakes considered in Table 5 which are closer than 75 km to each one of the sites.

Similar results are obtained for all three locations for the considered models regarding the monthly variations (Figure 14). Overall, the monthly variations do not show changes preceding relevant earthquakes for this case study. One of the possible explanations is the lack of foreshocks in most of the main shocks. In Lorca earthquake, even though there was a 4.5 Mw earthquake almost two hours before the main-shock, the one-month increments on the computation process are not able to show any change in RC.

The annual variations (Figure 15), on the other hand, show periods of increased RC before some of the selected earthquakes. An example is seen in Lorca site where a 15% increase is seen before Mula earthquake from June 1998 (the earthquake occurred in February 1999). Another example can be seen in both Murcia and Lorca sites, where a 10% increase can be seen before Aledo earthquake from May 2004 until the earthquake occurrence in January 2005. For this metric, differences between the three models can be seen. For instance, Model 1t and Model 2t show greater changes after Lorca's earthquake in Lorca's site.

The most prominent increase on the annual variation occurs after Lorca earthquake (32.8%, 36.2% and 21% for models 1t, 2t and 3t).

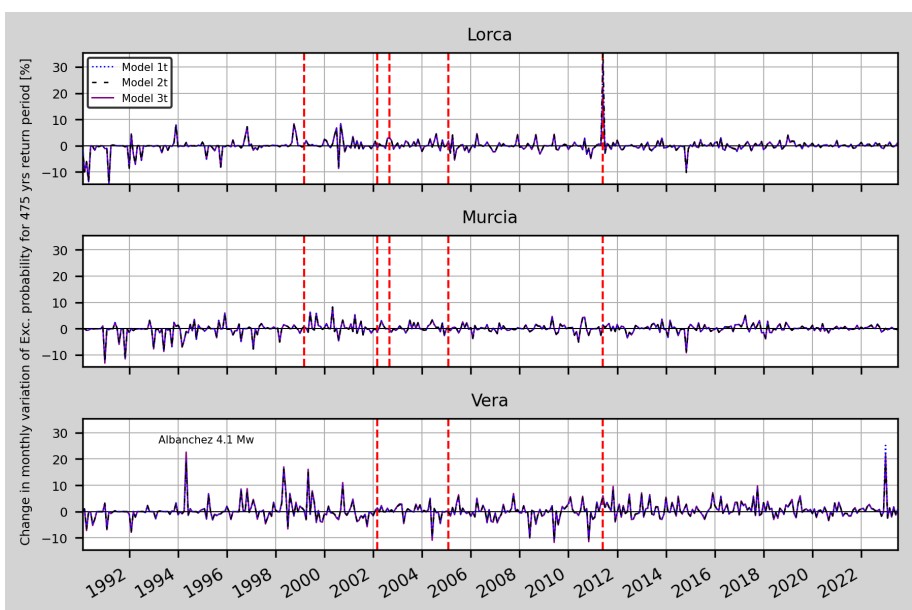

**Figure 14.** Monthly variations of the relative change of the annual probability of exceedance for models 1t, 2t and 3t in Lorca, Murcia and Vera (from top to bottom). The vertical dashed red lines mark the earthquakes considered in Table 5 which are closer than 75 km to each one of the sites. The earthquake that could cause the peak in 1994 at Vera site has also been indicated.

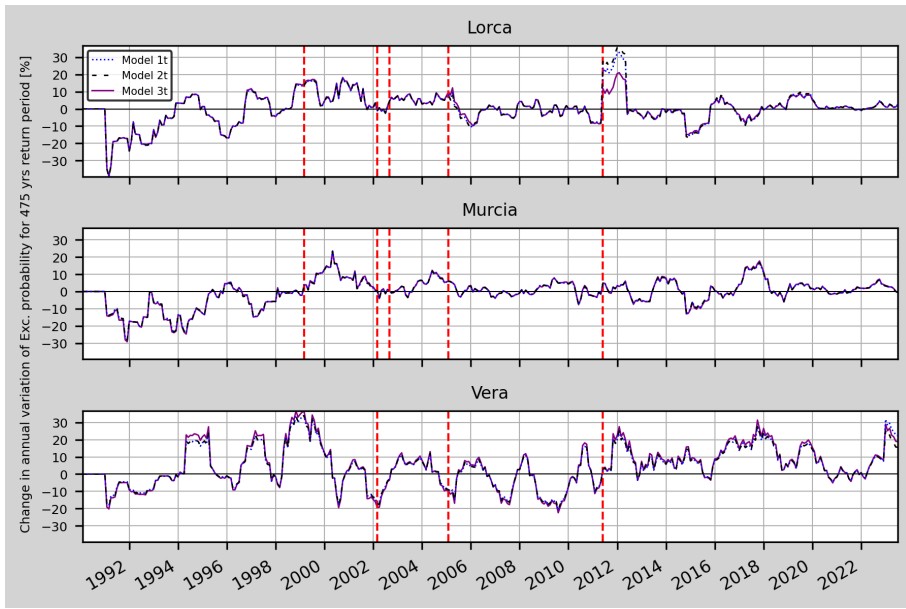

**Figure 15.** Annual variations of the relative change of the annual probability of exceedance for models 1t, 2t and 3t in Lorca, Murcia and Vera (from top to bottom). The vertical dashed red lines mark the earthquakes considered in Table 5 which are closer than 75 km to each one of the sites.

□ *Effect of the declustering on the results*

In order to compare the effect of the catalogue declustering on the results, the Model 1t has been plotted using both the declustered catalogue (with a total of 13841 events) and the full catalogue (with the clusters identified and weighted down accordingly).

  Figure 16 represents the changes in the annual exceedance probability for Model 1t. As can be seen, the results using a non-declustered catalogue provide lower changes in the exceedance probability for Mw 4.0 in Lorca site from 1996 until 2011.

Then, from 2011 until 2023, the non-declustered catalogue provided a higher RC. At both Murcia and Vera, the results are similar for the non-declustered and declustered catalogues. It should also be noted that the mean uncertainty of the RC is slightly higher for the declustered catalogue (a 11.21% for Lorca, -0.41% for Murcia and 5.58% for Vera). Since the results are compatible, keeping the foreshocks and aftershocks, i.e. using the non-declustered catalogue, seems to be a better choice if the aim is to perform OEF. Some of the advantages would be a lower uncertainty in the RC and the possibility of using more

detailed time scales in case foreshocks are present.

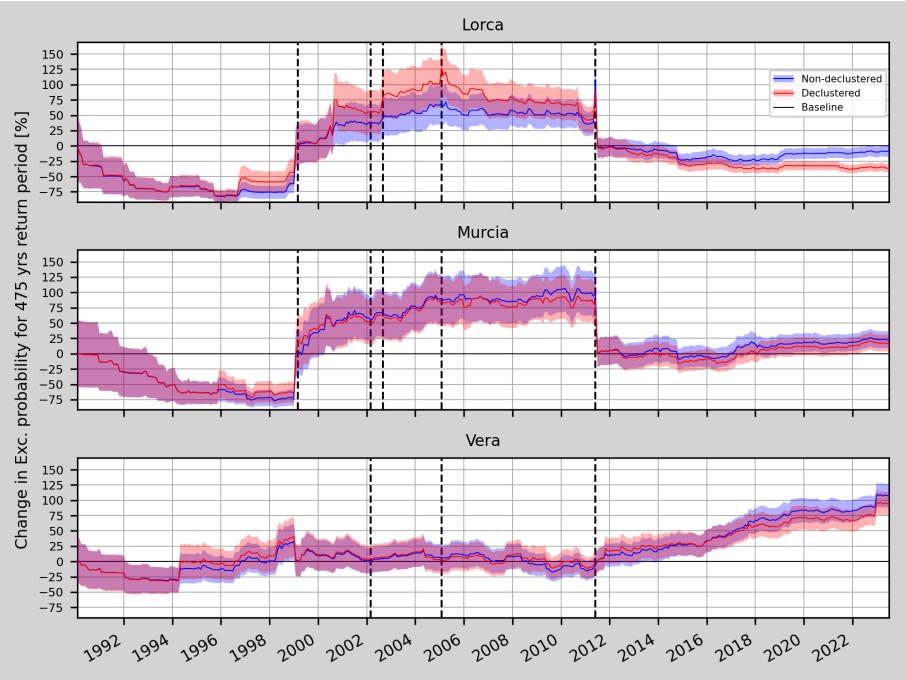

**Figure 16.** Model 1t. Comparison of the relative change (RC) of the annual exceedance probability and its uncertainty for a non-declustered and a declustered catalogue in Lorca, Murcia and Vera (from top to bottom). The vertical dashed black lines mark the earthquakes considered in Table 5 which are closer than 75 km to each one of the sites.

## 4 Conclusions

This methodology considers the influence of all the events in the seismic clusters and also the location of the seismic sources (corresponding active faults) for seismic activity rate smoothing and b-value computation, showing that when computing a time-dependent PSHA the use of a non-declustered catalogue will provide similar results to using a declustered catalogue with the added benefit of keeping the foreshock activity. Therefore, if we compute the changes of the annual probability of exceedance for a given PGA value (fixed as a background value which may change according to the updates in the seismic normative), we will be able to show how this probability is changing with time.

The changes in the annual probability of exceedance (increases and decreases) can be more accurately described using a spatially gridded time-dependent b-value instead of a fixed one for each tectonic zone. This can be seen when comparing Figure 10 with Figure 12. Therefore, we suggest using spatially gridded b-values for the corresponding period (time-dependent) when computing the background PGA value and the corresponding changes in the annual probability of exceedance in the time-dependent PSHA.

Regarding which of the proposed models can be more effectively used to describe these changes, we have to consider several factors. One could be how close are the computed PGA values to the national seismic hazard maps for each country. In the case of Central Italy models 1t, 2t and 3t provide the following background PGA values: 340.61, 359.72 and 334.28 $\mathrm{cm/s^2}$. The ESHM20 model (Danciu et al., 2021) computes 334.38 $\mathrm{cm/s^2}$. The closest match would be Model 3t followed by Model 1t. However, by looking at Figure 5 and Figure 7, it can be seen that the Model 3t seems to be less affected by changes in the seismic activity than Model 1t, as the monthly and annual RC variations suggest (Model 1t monthly and annual variations are 4.5 times higher than Model 3t variations). With this information Model 1t seems appropriate for the purpose of this work.

In general, this methodology benefits from complete catalogues in zones with increased seismicity - assuring less uncertainty in the b-value computation - and well-defined seismicity sources, where the seismicity smoothing is accurate. Figure 16 shows this result, as the non-declustered catalogue (with weighted down cluster events) has less RC uncertainty and enables the use of the foreshocks in daily to weekly time scales.

Although our results are not significant to relate these changes to the occurrence of a main earthquake for low to moderate seismicity areas, the methodology can be useful for other countries with a higher seismicity, or in the future if new significant earthquakes occur in the studied region of Spain. As we saw, for Central Italy both the annual and monthly changes of the exceedance probability show important variations related to the foreshock activity preceding L'Aquila earthquake. This could be useful for OEF.

Finally, in the case of south-eastern Spain, the relative change in the annual probability of exceedance kept high in the region after Mula earthquake and did not decrease until the occurrence of the Lorca earthquake. However, the continuous increase in this parameter in Vera after the Lorca earthquake cannot be directly related to a potential upcoming earthquake similar to the one from Lorca. Therefore, more time and data are needed to confirm this.

*Code availability.* The code used in this study is available upon request.

*Data availability.* The catalogues used in this work can be found in the following repository: https://doi.org/10.5281/zenodo.13691624. The

original catalogues can be found in Lolli et al. (2020) for Italy and in Instituto Geográfico Nacional (IGN) for Spain.

## Appendix A:  Other results for South-eastern Spain

### A1    *Time-dependent PSHA using time-dependent b-value (time-dependent model)*

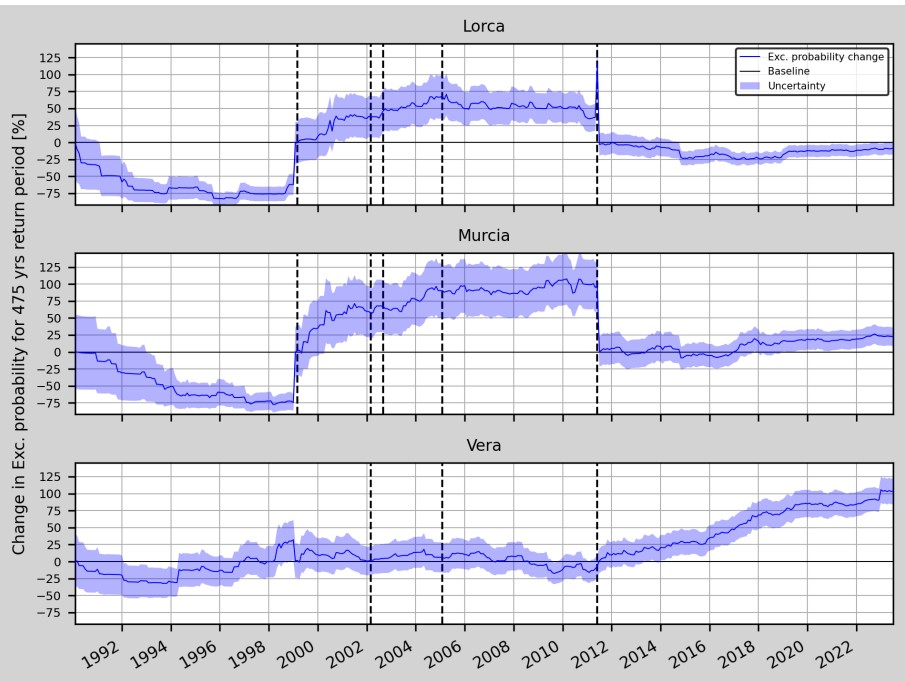

**Figure A1.** Relative change (RC) of the annual exceedance probability and corresponding uncertainty for Model 2t in Lorca, Murcia and Vera (from top to bottom). The vertical dashed black lines mark the earthquakes considered in Table 5 which are closer than 75 km to each one of the sites.

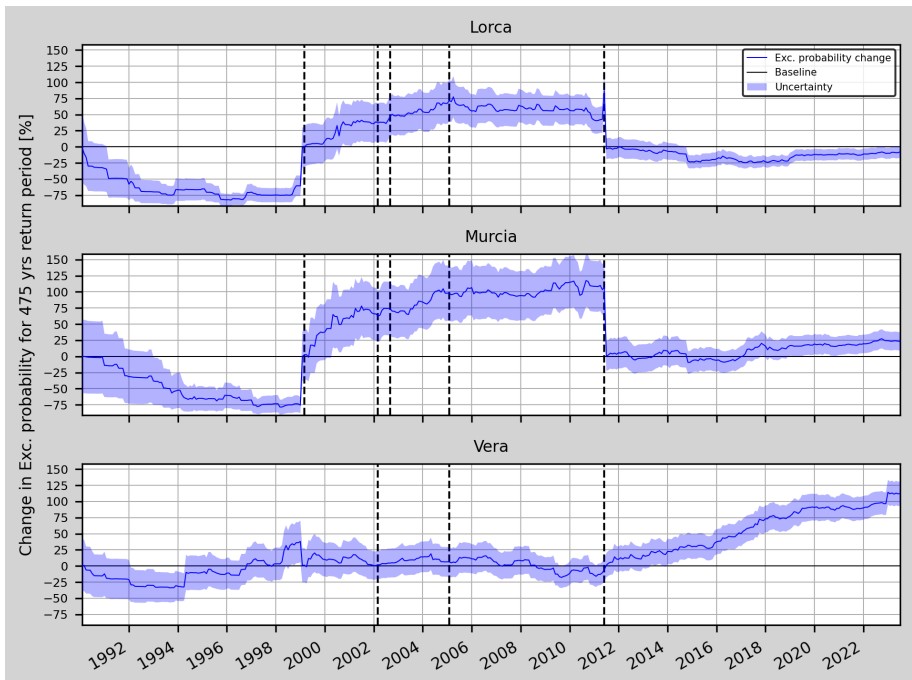

**Figure A2.** Relative change (RC) of the annual exceedance probability and corresponding uncertainty for Model 3t in Lorca, Murcia and Vera (from top to bottom). The vertical dashed black lines mark the earthquakes considered in Table 5 which are closer than 75 km to each one of the sites.

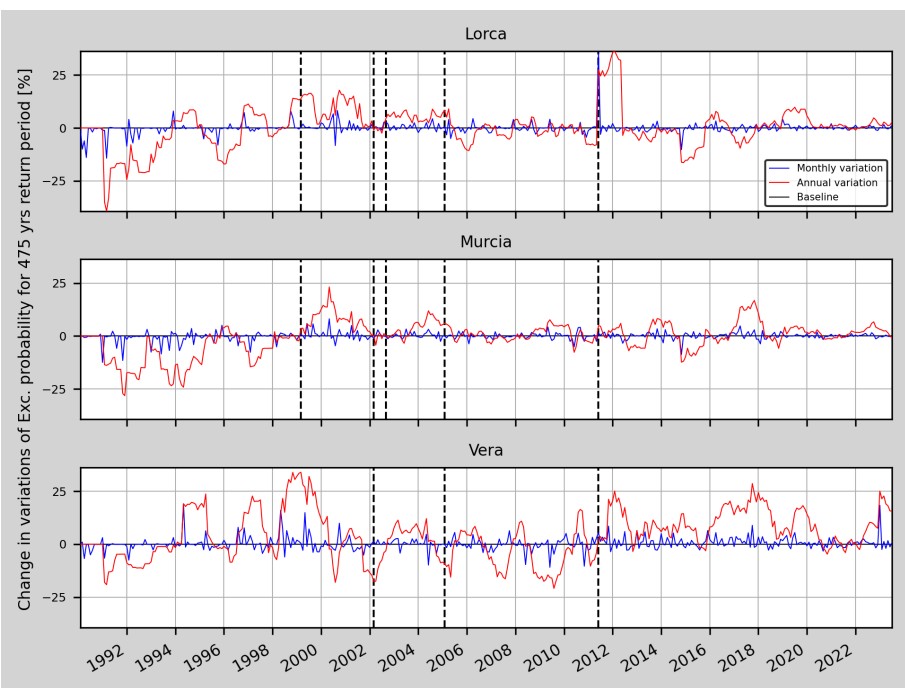

**Figure A3.** Annual and monthly variations of the relative change of the annual probability of exceedance for Model 2t in Lorca, Murcia and Vera (from top to bottom). The vertical dashed black lines mark the earthquakes considered in Table 5 which are closer than 75 km to each one of the sites.

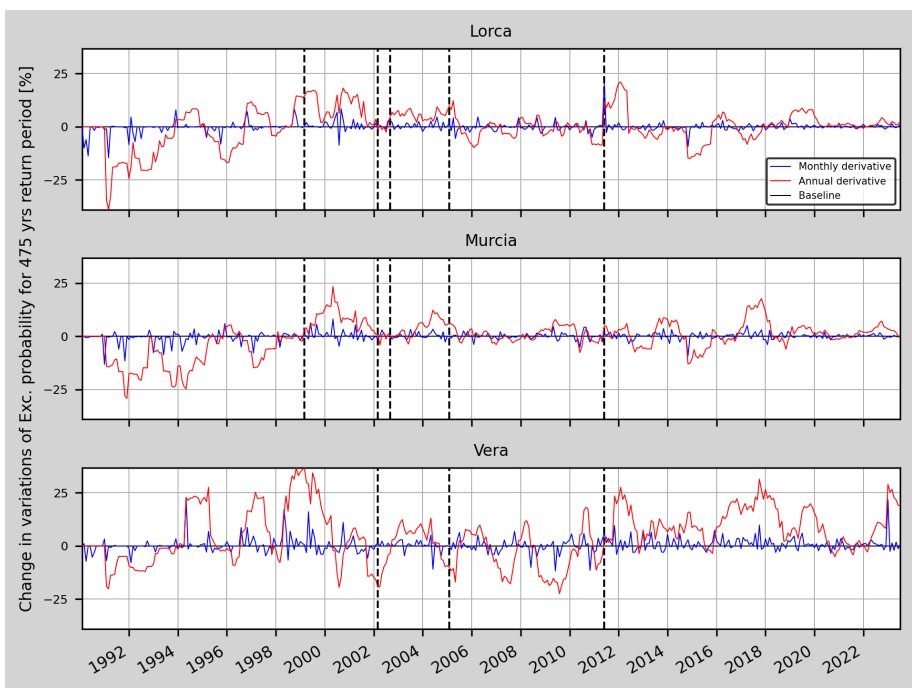

**Figure A4.** Annual and monthly variations of the relative change of the annual probability of exceedance for Model 3t in Lorca, Murcia and Vera (from top to bottom). The vertical dashed black lines mark the earthquakes considered in Table 5 which are closer than 75 km to each one of the sites.

## A2  *Effect of the declustering on the results*

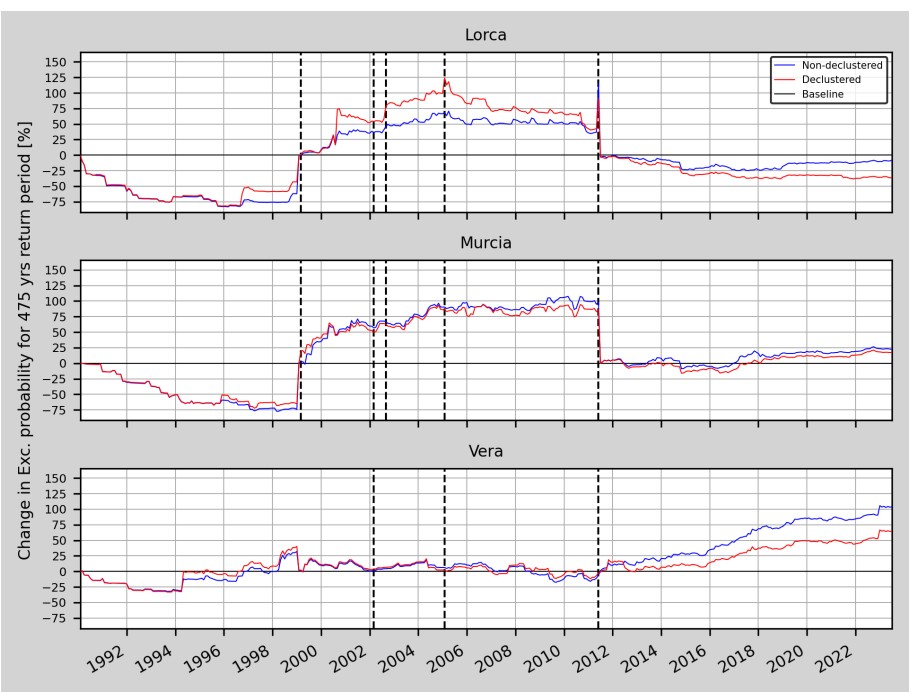

**Figure A5.** Model 2t. Relative change (RC) of the annual exceedance probability for a non-declustered and a declustered catalogue in Lorca, Murcia and Vera (from top to bottom). The vertical dashed black lines mark the earthquakes considered in Table 5 which are closer than 75 km to each one of the sites.

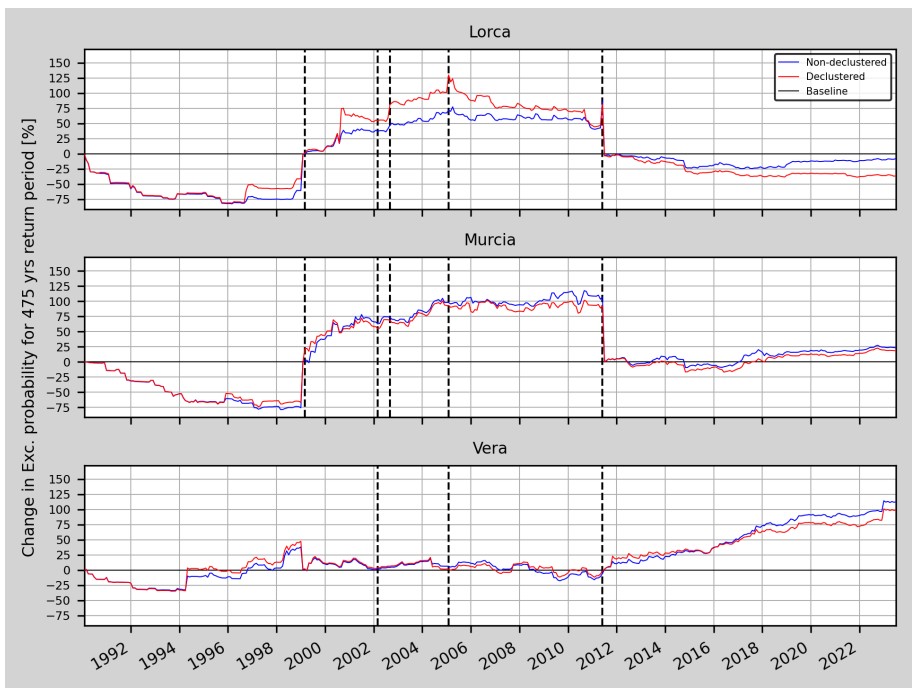

**Figure A6.** Model 3t. Relative change (RC) of the annual exceedance probability for a non-declustered and a declustered catalogue in Lorca, Murcia and Vera (from top to bottom). The vertical dashed black lines mark the earthquakes considered in Table 5 which are closer than 75 km to each one of the sites.

*Author contributions.* Conceptualisation and original idea: DML and SM; methodology: DML, SM, JJGM, IG, AK, JLSL, JAHT, AGV and GOS; DML wrote the code and tested the different components; DML and SM performed the data curation; writing the original draft: DML, SM and JJGM; writing review and editing: DML, SM, JJGM, IG, AK, JLSL, JAHT, AGV and GOS. All authors have read and agreed to the published version of the manuscript.

*Competing interests.* The authors declare that they have no conflict of interest.

*Acknowledgements.* This research was partially supported by the European Regional Development Fund (FEDER), the Government of Spain (Ministry of Science and Innovation) through the reference project PID2021-123135OB-C21, by the Regional Government of Valencia (Ministry of Education, Culture, Universities and Employment) through the reference project: CIAICO/2022/038, and by the Research Group VIGROB-116 (University of Alicante).

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
