# Peer review of "Computing time-dependent activity rate using non-declustered and declustered catalogues. A first step towards time-dependent seismic hazard calculations for operational earthquake forecasting"

_EGUsphere, 2023_

## Referee Comment (RC1)

[referee-annotated manuscript omitted]

---

## Author Comment (AC1)

We appreciate the referee's comments as they were both helpful and enlightening. All the highlighted typos have been corrected but not addressed in this document in order not to overextend the response. Only major changes and clarification notes are presented. The numeration has changed and the figures' and tables' numeration refers to the last version of the manuscript.

**Page 1: Abstract.**

**Do you mean the distance from the centre of a square cell? The same in the following when you speak about distance from a cell.**

The abstract has been modified in order to use a more concise language: every reference to the cell distance has now been regarded as distance to the centre of the grid cell to avoid any ambiguity (p.1 line 5, p.3 line 58, etc.).

**The word "oscillation" refers to alternate episodes of increase and decrease of the probability, but it does not seem appropriate to the results shown in this paper.**

Regarding the use of the word oscillation, we agree that it is no appropriate in this context so the whole sentence has been changed:

Lines 14-17:

"*In the case of Italy, the annual probability of exceedance increases significantly, but in the case of Spain not all the earthquakes have an associated increase in the exceedance probability.*"

**Page 5: 2.1 Smoothing kernel**

**Please, give some references.**

**This function does not seem a 2D Gaussian function. The maximum is not in a point, but on a circle of radius "mu".**

Lines 115-116:

The description of the smoothing kernel has been changed to correctly describe its effects and a citation to a similar function has been added:

"*For this work, a well-known smoothing function (Frankel, 1995) has been selected to smooth the gridded seismicity, the 1D Gaussian function (Eq. 6):*"

When the distance from the point in which the seismic activity rate is being computed to the centre of the spatial grid cell that contributes to that computation is calculated, the variable of interest becomes 1D rather than 2D (if the longitude and latitude of each point where considered) and the expression of the smoothing function becomes simpler.

Lines 119-121:

The description of the smoothing kernel has been changed to the following:

$$f(r) = A \cdot \exp\left(-\frac{(r - \mu)^2}{2 \cdot \sigma^2}\right)$$

*"where **r** is the distance between the centre of the spatial grid cells and the centre of the cell in which the seismic activity is being computed, **A** is the normalization constant, **μ** is the parameter that controls the **r** value at which the maximum of the function is reached, and **σ** controls the dispersion of the function around the maximum value."*

This has been done to avoid a direct relationship between the parameters on the smoothing function to those of the normal distribution PDF before arriving to the discussion section.

Additionally, a new subsection has been created to illustrate the difference between the $\mu = 0$ and $\mu = d_{f_i}$ cases, as well as an example of code implementation.

==There is confusion between the mean and the modal value (maximum of the distribution).==

- **First moment of the distribution, μ** has been renamed to **Geophysical meaning of the parameter μ** and the contents of the subsection have been rewritten in order for them to be more clearly defined.

*"The meaning of this parameter within the context of the seismic activity smoothing for this model is the distance from a given cell centre to the point(s) in which the probability of having an earthquake is higher.*

*It is common to find that the value of this parameter is set to zero **(Frankel, 1995; Helmstetter et al., 2006; Hiemer et al., 2014)**, as the maximum probability of having an earthquake is where it has already happened before. So, the smoothing function has its maximum value in the cell in which the seismic activity rate is being smoothed. This constitutes the first option regarding this parameter: **μ = 0**.*

*An alternative model is proposed, where the maximum probability is set at the location of the nearest seismic sources. For this to be implemented, the minimum distance between the point in which seismic activity rate is being computed and the location of the nearest seismic source is calculated and named in this work from now on as $d_{f_i}$. So, the second option for the parameter value is $\mu = d_{f_i}$.*

*For areas in which the tectonic structures are only present in part of the region, a hybrid approach may be applied by using cut-off distance. This cut-off distance may be calculated as follows:*

$$d_c = \bar{d} + 2 \cdot \sigma_d$$

*where $d_c$ is the cut-off distance, $\bar{d}$ stands for the mean value of the distance between all the structures, and $\sigma_d$ is the standard deviation for all these distances.*

*If the distance from the centre of the spatial grid cell to the nearest fault is higher than the cut-off distance then $\mu = 0$. Otherwise, it will be set to $d_{f_i}$*"

**It does not seem to me that it is the dominant factor of the dispersion. There is also a physical dispersion.**

- **Second moment of the distribution, σ** has been renamed to **Geophysical meaning of the parameter σ** and the contents of the subsection have been rewritten in order for them to be more clearly defined.

*"This parameter accounts for the dispersion of the values of the distribution around the mean value. That is to say, how far one might expect to find earthquakes around the most probable value (of distance). Therefore, we have considered that this second parameter is related to the accuracy of earthquake's epicentre measurement. This means that it would depend on the methodologies and instrumentation used for the calculation of the epicentre, and thus, on both the year and the location of the catalogue.*

*It should be noted that **σ** may depend on other geophysical parameters such as the characteristics of ground, the style of faulting and/or the tectonic stress regime, to cite a few. Nevertheless, in this work only the influence of the uncertainty in the epicentre's location will be considered in the smoothing process.*

*As in the previous section, two different options regarding the epicentre uncertainty, **ε**, have been considered: either it depends on the year of occurrence (**ε₁**), or it is constant and computed as the mean value of the epicentral uncertainty for all the events (**ε₂**). "*

A new subsection has been added to show examples of the smoothing kernel implementation and a figure showcasing how the smoothing works is presented:

**2.1.3 Examples of the smoothing kernel implementation**

*"In this section, some examples of how the smoothing kernels works are shown. There are three main different manners in which this smoothing is applied:*

- *Usual 1D Gaussian filter, $\mu = 0$*

*This is the case when using models 2 and 3 also when the distance from the centre of the spatial grid cell in which the seismic activity rate is computed to the nearest fault is greater than $d_c$ as defined in the section **2.1.1.** An example can be seen in **Figure 1a.***

- *Single fault, $\mu \neq 0$*

*When the nearest fault is closer than $d_c$ from the centre of the spatial grid cell then the resulting function will provide a ring-shaped smoothed activity, the width of which will depend on **σ**. Only the section of this ring in which the fault is located will be used in the smoothing. This can be achieved by considering the **n** closest points*

to the spatial grid cell centre and then computing the angles to define the ring arc (*Figure 1b*).

- Several faults, $\mu \neq 0$

This case is a generalization of the former with the exception that when spatial grid cell's centre is in between faults and at similar distances, then the full ring will be used as smoothing function (*Figure 1d*). On the other hand, if the distance to both faults is similar, but the spatial grid cell's centre is not in between the faults then the resulting smoothing is a ring arc (*Figure 1c*)."

[Figure]

*Figure 1. a) Smoothing function for $\mu = 0$. b) Smoothing function for $\mu \neq 0$ and a single fault. c) Smoothing function for $\mu \neq 0$ and several faults at similar distances. d) Smoothing function for $\mu \neq 0$ and the spatial grid cell in between faults at similar distances. The blue lines show the fault traces. In this example $d_c$ equals 48 km.*

Statistical methods were introduced for declustering without removing events from a catalogue, but assigning to each earthquake a weight equal to the probability of being independent These methods make use of the Epidemic (ETAS) model. See, e.g.:

ZHUANG, J., OGATA, Y., and VERE-JONES, D. (2002), Stochastic declustering of space-time earthquake occurrences, J. Am. Stat. Ass. 97, 458, 369–380.

MARSAN, D. and LONGLINE´, O. (2008), Extending earthquakes' reach through cascading, Science 319, 1076–1079.

Console, R., Jackson, D.D., Kagan, Y.Y. (2010). Using the ETAS model for catalog declustering and seismic background assessment. Pure Appl. Geoph., 10.1007/s00024-010-0065-5.

These algorithms, based on parameters whose value is established by the maximum likelihood, avoid the difference between declustered catalogues obtained from a subjective choice of different algorithms.

**Page 8: 2.2 Cluster identification and seismicity smoothing**

A paragraph has been introduced discussing the methodology and taking into account the references.

*"First, the spatial grid is defined by creating a rectangle spanning the maximum and minimum longitudes and latitudes of the catalogue with the desired resolution. Then, all the events of the catalogue must be assigned to each cell. This is done by calculating the minimum distance of each event to all the spatial grid cell's centres.*

*One of the most important steps regarding the activity rate calculation in this work, is the identification of the seismic clusters present in the area for the selected period of time. As indicated in the introduction, we do not pretend to remove the foreshocks and aftershocks but to identify the main event and all related events in the corresponding cluster.*

*To do so, even though Epidemic Type Aftershock Sequence (ETAS) model allows to assign to each event the probability of being an aftershock (Zhuang et al., 2002; Marsan and Longline, 2008; Console et al., 2010) in this work we have decided to select a non-stochastic method based on the performance classifying events of a relevant seismic series."*

Additionally, the procedure in which the methods are tested has been expanded by computing the confusion matrix for all the methods given the knowledge on the Lorca's series. Now after **Table 6** in **page 19** is presented:

*"Considering that Cabañas et al. (2011) carried out a detailed study on the 2011 Lorca's earthquake seismic series, we have used their results to validate the best algorithm. According to them, the cluster corresponding to Lorca's series, from 11 May 2011 until 19 July 2011, is composed of 146 events (including the foreshock, the main shock and the aftershocks). In order to test the performance of the declustering methods, the confusion matrices for each one have been computed. In the area of study, a total of 249 events have been recorded, which means a total of 103 background events should be identified. For this analysis, all the events classified in a cluster different from Lorca series' have been considered as*

*background events for simplicity. **Figure 9** shows that GK74 method is the most adequate (with a 94.43% mean for the metrics compared with the 92.88% for RJ and a 74.54% for A) and also the one that is able to identify more events belonging to Lorca series."*

[Figure]

*Figure 9. Confusion matrices for the tested declustering methods. Inside each square, the number of events (bold) and some metrics computed using the data are presented (NPV stands for Negative Predictive Value).*

**The maps of this figure should be redrawn with a larger size.**

**Page 11: Figure 3**

The maps in the **Figure 3** have been resized so the features can be clearly seen. And the size in the draft has also been increased from 8.3 cm to 12 cm. It has been checked that the legend items can be read at 100% zoom value in the pdf version of the draft and other text processing software.

[Figure]

Too long sentence.

**Page 10: 2.4. Exceedance probability calculation**

Lines 219-224:

We agree that the sentence is too long, and it has been rewritten:

*Our goal is to investigate if the changes in the seismic activity and b-value time series can be observed as a trend in the PSHA results. In the case such trends are observed, this methodology could be used for OEF. For this reason, the temporal evolution of the annual probability of exceedance (PoE) of a background PGA corresponding to a 475 years return period (i.e., 0.002 PoE) has been computed as a time-dependent value. The results have been expressed as a relative change (RC in percentage change, Eq. 9) between background annual exceedance probability (long-term value) and the time-dependent annual exceedance probability.*

Do you mean that the background value is updated in time by the seismic normative?

Line 224-227:

The sentence has been rewritten to avoid any ambiguity and placed just before *Eq. 9* as it is related to the background value used to calculate the RC:

*Depending on the country, the background (long-term) PGA value may have been updated in the corresponding seismic hazard studies. This could be due to the*

*occurrence of new damaging earthquakes or improvements in the seismic knowledge of a region, amongst other reasons. In case such changes have been made, a new background PGA value has been computed using the data up until the year the seismic hazard information was updated.*

**It is well known that foreshock activity occurred in the weeks prior to the Aquila earthquake, with a notable increase just few days before. It is important to clarify if the increase in the exceedance probability is caused by this foreshock activity.**

**Page 14: 3.1.2 Results**

Lines 260-266:

In the case of the L'Aquila earthquake the foreshock activity allows to see clear changes in PoE as the earthquake preceding the mainshock has magnitude greater than Mw 4.0. That is one of the reasons why we decided to weight down the not independent events rather than eliminating them from the catalogue. The sentence has been rewritten in order to relate the changes in the RC prior the L'Aquila earthquake to the foreshock activity:

*"**Figure 5** (Model 1t) presents a moderate increase in the annual exceedance probability (25%) one month before L'Aquila earthquake occurred, and not only the annual but also the monthly variation of relative change reaches values higher than 35%. This sudden change is most probably due to the foreshock activity that preceded the mainshock, as a 4.1 $M_L$ ground motion occurred on 30 March 2009."*

**Table 5 reports only five earthquakes of the seven. Why?**

**Page 16: 3.2.1 Catalogue preparation and parameters for computation**

Lines 276-279:

The sentence has been rewritten in order to explain the choice of the 5 earthquakes:

*"Additionally, in the last 25 years, South-eastern Spain has suffered seven earthquakes with Mw greater than or equal to 4.5 (**Table 5** and **Figure 8** present only those classified as mainshocks and located in the area of study), being the 2011 Lorca earthquake the most relevant since it was the most recent earthquake causing damage to buildings and injuries to the population."*

**Page 13: Figure 4**

Caption line 2:

The word *tectonic* has been omitted as it is not correct. It was a reference to the tectonics' zone b-value, which is redundant and can induce misunderstanding. All the sentences in which the adjective *tectonic* precedes b-value have been omitted.

**Better with respect to which criterion?**

**Page 21: 3.2.2 Results**

Lines 332-336:

The sentence has been rewritten as it was neither accurate nor explanatory. The final sentence refers to the figures for the models 2t and 3t (as the one presented in *Figure 12* is the comparison between all the models).

*"After computing the time-dependent PSHA for the different models shown in **Table 9**, we have observed that although all the graphs show similar behaviour for the RC, Model 1t provides greater annual and monthly variations of the RC for some of the earthquakes than the rest, similarly to Italy's case study. Therefore, with the exception of the fixed models (models 1f, 2f and 3f) where we present a general comparison them, we will present the results of Model 1t in this section along with figures comparing all three models. The stand-alone figures for the models 2t and 3t can be found in the **Appendix** section."*

**Please, explain better how you justify this statement.**

**Page 22: 3.2.2 Results**

Lines 357-362:

The paragraph has been modified with a further explanation. The results presented in **Figure 10**, the comment of which is presented in this paragraph, will also be related to the ones from the time-dependent models (1t, 2t and 3t) in the conclusions.

*"It seems that the use of a constant b-value coupled with a time-dependent seismic activity rate leads to a RC that decrease over time with a constant rate and sudden increases mainly due to changes in the background PGA values. This behaviour is due to the update of this parameter depending on the period of the catalogue, which is increased a month at a time (as it was the selected minimum time step for this area). We find that this approach is not appropriate towards earthquake forecasting for areas with low to moderate seismicity. This uniform behaviour potentially rules out the possibility of finding any metrics for OEF.*

**It is important to distinguish if the sharp increase includes the data of the Mula earthquake or it includes only data before the earthquake. It makes a big difference, which must be clarified.**

Lines 364-377:

Mula earthquake occurs after the RC increase mentioned in the manuscript. Nevertheless, this has been corrected as this change is not the one to be addressed when discussing the results. The change that now appears in the discussion ranges from -75% to -60% that is seen from June 1998 on. The sudden change seen from December 1998 to January 1999 is an artifact due to the base change. This has been now explained and it is the reason why two other metrics have been selected

(monthly and annual variations). This is not the case for the Italian catalogue, as only one background PGA value has been used (at the start of the study period). The paragraph has been rewritten to account for this explanation:

"*In this section, models 1t, 2t and 3t (**Table 9**) are tested using the three PGA background values explained previously (and computed in January 1990, December 1998 and May 2011). As it can be seen in **Figure 11** the annual probabilities decrease before Mula earthquake for Lorca site. However, close to the occurrence of the earthquake, it shows a slight increase even in Vera site, although it is 101.4 km away from the earthquake's epicentre. In Murcia site, the RC continuously decreases until five months before the earthquake, when it shows a sharp increase from -75% to -60% in the change of exceedance probability. This change is also seen in the annual and monthly variations of the RC (**Figure 11 zoom in**).  After the Mula earthquake the change in probability exceedance remains higher than 20% (even increasing up until 50% in the case of Lorca and 100% in the case of Murcia) for both Lorca and Murcia sites until Lorca earthquake happens. In Vera site, this parameter oscillates about the baseline. After 2011, the RC steadily increases in Vera, whereas in Lorca and Murcia it stays constant after 2019. On the other hand, Model 2t and Model 3t (**Figure 12**) show a similar behaviour to Model 1t, although Model 3t showcases slightly higher exceedance probability before Lorca 2011 earthquake for both Murcia and Lorca sites. It should be noted that the sudden changes in the RC in January 1999 and one month after the Lorca earthquake, i. e., the -60% to 0% increase in January 1999 (for Lorca and Murcia) and the 100% to 0% decrease in June 2011 (for both Lorca and Murcia), are artefacts due to the change in the background PGA and cannot be considered in the analysis. However, the annual* and *monthly variations of the RC shown in **Figure 13** and **Figure 14** allow to see the changes related to the aforementioned earthquakes occurrences.*"

As for the differences in the models we agree that they are similar in the behaviour and just small changes can be appreciated in some periods. The main differences can be observed in the annual variations of the RC. The **Figure 11** has been updated to illustrate the discussion of the results by including a zoom in.

**What about the sharp decrease in Lorca and Murcia after the Lorca earthquake?**

"*It should be noted that the sudden changes in the RC in January 1999 and one month after the Lorca earthquake, i. e., the -60% to 0% increase in January 1999 (for Lorca and Murcia) and the 100% to 0% decrease in June 2011 (for both Lorca and Murcia), are artefacts due to the change in the background PGA and cannot be considered in the analysis.*"

[Figure]

*Figure 11. Relative change (RC) of the annual exceedance probability and corresponding uncertainty for Model 1t in Lorca, Murcia and Vera (from top to bottom). The vertical dashed black lines mark the earthquakes considered in Table 5 which are closer than 75 km to each one of the sites. A zoom in on the mentioned increase in the RC during 1998 in Lorca's site appears in the upper left side of the graph.*

The symbology in **Figure 12** has been changed so that the Model 3t is better seen.

[Figure]

*Figure 12. Mean value of the relative change (RC) of the annual exceedance probability for models 1t, 2t and 3t in Lorca, Murcia and Vera (from top to bottom). At the right side, a zoom in on the peak due to Lorca earthquake. The vertical dashed red lines mark the earthquakes considered in Table 5 which are closer than 75 km to each one of the sites.*

**Page 24: Comment on Figure 13**

The paragraph has been rewritten as the **Figure 13** has been modified due to a change in the representation of the data that affected several points in the graph. Lines 378-389:

**Is this change significant of just a random variation?**

[revised manuscript text omitted]

**Page 28-29: Conclusions.**

The conclusions have been rewritten:

"*This methodology considers the influence of all the events in the seismic clusters and also the location of the seismic sources (corresponding active faults) for seismic activity rate smoothing and b-value computation, showing that when computing a time-dependent PSHA the use of a non-declustered catalogue will provide similar results to using a declustered catalogue with the added benefit of keeping the foreshock activity. Therefore, if we compute the changes of the annual probability of exceedance for a given PGA value (fixed as a background value which may change according to the updates in the seismic normative), we will be able to show how this probability is changing with time.*

*The changes in the annual probability of exceedance (increases and decreases) can be more accurately described using a spatially gridded time-dependent b-value instead of a fixed one for each tectonic zone. This can be seen when comparing* **Figure 10** *with* **Figure 12**. *Therefore, we suggest using spatially gridded b-values for the corresponding period (time-dependent) when computing the background PGA value and the corresponding changes in the annual probability of exceedance in the time-dependent PSHA.*

*Regarding which of the proposed models can be more effectively used to describe these changes, we have to consider several factors. One could be how close are the computed PGA values to the national seismic hazard maps for each country. In the case of Central Italy models 1t, 2t and 3t provide the following background PGA values: 340.61, 359.72 and 334.28 cm/s$^2$. The ESHM20 model (Danciu et al., 2021) computes 334.38 cm/s$^2$. The closest match would be Model 3t followed by Model 1t. However, by looking at **Figure 5** and **Figure 7**, it can be seen that the Model 3t seems to be less affected by changes in the seismic activity than Model 1t, as the monthly and annual RC variations suggest (Model 1t monthly and annual variations are 4.5 times higher than Model 3t variations'). With this information Model 1t seems appropriate for the purpose of this work.*

*In general, this methodology benefits from complete catalogues in zones with increased seismicity - assuring less uncertainty in the b-value computation - and well-defined seismicity sources, where the seismicity smoothing is accurate. **Figure 16** shows this result, as the non-declustered catalogue (with weighted down cluster events) has less RC uncertainty and enables the use of the foreshocks in daily to weekly time scales."*

==Probably due to the foreshock period before the mainshock.==

*Although our results are not significant to relate these changes to the occurrence of a main earthquake for low to moderate seismicity areas, the methodology can be useful for other countries with a higher seismicity, or in the future if new significant earthquakes occur in the studied region of Spain. **As we saw, for Central Italy both the annual and monthly changes of the exceedance probability show important variations related to the foreshock activity preceding L'Aquila earthquake. This could be useful for OEF**.*

*Finally, in the case of south-eastern Spain, the PSHA kept high in the region after the Mula earthquake and did not decrease until the occurrence of the Lorca earthquake. However, the continuous increase of the PSHA in Vera after the Lorca earthquake cannot be directly related to a potential upcoming earthquake similar to the one from Lorca. Therefore, more time and data are needed to confirm this."*

==A better interpretation of the results is needed. Why sometimes the PSHA increase after a mainshock and other times it decreases suddenly?==

Response in pages 8 and 9 of this document.

**Page 34-35: Appendix A2 Effect of the declustering on the results**

**Figure A5** has been corrected as Lorca's site data was erroneous.

[Figure]

*Figure A5. Model 2t. Relative change (RC) of the annual exceedance probability for a non-declustered and a declustered catalogue in Lorca, Murcia and Vera (from top to bottom).*

**References added:**

*Hiemer, S., Woessner, J., Basili, R., Danciu, L., Giardini, D., and Wiemer, S.: A smoothed stochastic earthquake rate model considering seismicity and fault moment release for Europe, Geophysical Journal International, 198, 1159-1172, https://doi.org/10.1093/gji/ggu186, 2014*

*Zhuang, J., Ogata, Y., and Vere-Jones, D.: Stochastic declustering of space-time earthquake occurrences, J. Am. Stat. Ass. 97, 458, 369–380, 2002*

*Marsan, D. and Lengliné, O.: Extending earthquakes' reach through cascading, Science 319, 1076–1079, 2008.*

*Console, R., Jackson, D. D. and Kagan, Y. Y.: Using the ETAS model for catalog declustering and seismic background assessment. Pure Appl. Geoph., 10.1007/s00024-010-0065-5., 2010.*

---

## Author Comment (AC2)

First of all, thank you, Prof. Chan, for your comments and suggestions in the review of the manuscript. In the following document, all the questions will be answered in order and quoting each item of the review using the yellow highlighted and bold font style. The numeration has changed and the figures' and tables' numeration refers to the last version of the manuscript.

**The Italy case not discussed: The abstract and various sections of the manuscript mention the application of this approach to the L'Aquila, Italy, case. However, I am unable to find any corresponding results or discussion.**

The corresponding paragraphs (section **3.1.2 Results** and **4.0 Conclusions**) have been modified in order to discuss the results from Central Italy case.

**3.1.2 Results**

"*Figure 5 (Model 1t) presents a moderate increase in the annual exceedance probability (25%) one month before L'Aquila earthquake occurred, and not only the annual but also the monthly variation of relative change reaches values higher than 35%. This sudden change is most probably due to the foreshock activity that preceded the mainshock, as a 4.1 $M_L$ ground motion occurred on 30 March 2009. Figure 6 (Model 2t) shows a similar trend in all the metrics as the previous model with a slightly lower value for the exceedance probability change before the earthquake (22%) and the annual and monthly variations (32%). The Model 3t (Figure 7) provides the lowest values for the metrics (-3%, 4% and 3%, respectively). After this increase the RC slowly decreases over time for all three models.*"

**4.0 Conclusions**

*[...]*

"*Regarding which of the proposed models can be more effectively used to describe these changes, we have to consider several factors. One could be how close are the computed PGA values to the national seismic hazard maps for each country. In the case of Central Italy models 1t, 2t and 3t provide the following background PGA values: 340.61, 359.72 and 334.28 cm/s$^2$. The ESHM20 model (Danciu et al., 2021) computes 334.38 cm/s$^2$. The closest match would be Model 3t followed by Model 1t. However, by looking at Figure 5 and Figure 7, it can be seen that the Model 3t seems to be less responsive to the seismic activity than Model 1t, as the monthly and annual RC variations suggest (Model 1t monthly and annual variations are 4.5 times higher than Model 3t variations'). With this information Model 1t seems appropriate for the purpose of this work.*

*In general, this methodology benefits from complete catalogues in zones with increased seismicity - assuring less uncertainty in the b-value computation - and well-defined seismicity sources, where the seismicity smoothing is accurate. Figure 16 shows this result, as the non-declustered catalogue (with weighted down cluster events) has less RC uncertainty and enables the use of the foreshocks in daily to weekly time scales.*

*Although our results are not significant to relate these changes to the occurrence of a main earthquake for low to moderate seismicity areas, the methodology can be useful for other countries with a higher seismicity, or in the future if new significant earthquakes occur in the studied region of Spain. As we saw, for Central Italy both the annual and monthly changes of the exceedance probability show important variations related to the foreshock activity preceding L'Aquila earthquake. This could be useful for OEF."*

*[...]*

**Definition of smoothing kernel: In this study, the smoothing kernel is determined by the average distance between all events surrounding an earthquake and the precision of the epicenter's location (Lines 68-70). When defining the second moment of the distribution (Sec. 2.1.2), this parameter is solely attributed to the precision of the earthquake epicenter measurement. However, I anticipate that the distance between all events is equally important for this parameter.**

The smoothing kernel section and subsections have been rewritten in order to explain some of the features as well as to address this matter. Find here, the part of interest as well as the figure comparing the different cases (that arise from the different geometries and relations between faults). The rest of the modifications can be found in the modified version of the manuscript attached to the response.

- **Second moment of the distribution, σ** has been renamed to **Geophysical meaning of the parameter σ** and the contents of the subsection have been rewritten in order for them to be more clearly defined.

*"This parameter accounts for the dispersion of the values of the distribution around the mean value. That is to say, how far one might expect to find earthquakes around the most probable value (of distance). Therefore, we have considered that this second parameter is related to the accuracy of earthquake's epicentre measurement. This means that it would depend on the methodologies and instrumentation used for the calculation of the epicentre, and thus, on both the year and the location of the catalogue.*

***It should be noted that σ may depend on other geophysical parameters such as the characteristics of ground, the style of faulting and/or the tectonic stress regime, to cite a few. Nevertheless, in this work only the influence of the uncertainty in the epicentre's location will be considered in the smoothing process.***

*As in the previous section, two different options regarding the epicentre uncertainty, ε, have been considered: either it depends on the year of occurrence ($ε_1$), or it is constant and computed as the mean value of the epicentral uncertainty for all the events ($ε_2$). "*

A new subsection has been added to show examples of the smoothing kernel implementation and a figure showcasing how the smoothing works is presented:

**2.1.3 Examples of implementation**

*"In this section, some examples of how the smoothing kernels works are shown. There are three main different manners in which this smoothing is applied:*

- *Usual 1D Gaussian filter, $\mu = 0$*

*This is the case when using models 2 and 3 also when the distance from the centre of the spatial grid cell in which the seismic activity rate is computed to the nearest fault is greater than $d_c$ as defined in the section **2.1.1.** An example can be seen in **Figure 1a.***

- *Single fault, $\mu \neq 0$*

*When the nearest fault is closer than $d_c$ from the centre of the spatial grid cell then the resulting function will provide a ring-shaped smoothed activity, the width of which will depend on **σ**. Only the section of this ring in which the fault is located will be used in the smoothing. This can be achieved by considering the **n** closest points to the spatial grid cell centre and then computing the angles to define the ring arc (**Figure 1b**).*

- *Several faults, $\mu \neq 0$*

*This case is a generalization of the former with the exception that when spatial grid cell's centre is in between faults and at similar distances, then the full ring will be used as smoothing function (**Figure 1d**). On the other hand, if the distance to both faults is similar, but the spatial grid cell's centre is not in between the faults then the resulting smoothing is a ring arc (**Figure 1c**)."*

[Figure]

*Figure 1. a) Smoothing function for $\mu = 0$. b) Smoothing function for $\mu \neq 0$ and a single fault. c) Smoothing function for $\mu \neq 0$ and several faults at similar distances. d) Smoothing function for $\mu \neq 0$ and the spatial grid cell in between faults at similar distances. The blue lines show the fault traces. In this example $d_c$ equals 48 km.*

**Completeness magnitude: Based on my interpretation of Table 7, it appears that the magnitude range indicated in the top column has been complete from the year specified in the bottom column. Therefore, it should be that magnitudes of 3.0 and above are complete since 1978, rather than starting from magnitude 3.25 as stated in Lines 277-280. Additionally, the completeness magnitude typically decreases with upgrades to the seismic network, as such improvements generally enhance detection capabilities. It is customary for the completeness magnitude to remain stable for approximately a decade before decreasing sharply with a network upgrade. The gradual increase in the average completeness magnitude observed in Table 8 is unexpected. An explanation for the trend of decreasing completeness magnitude would be beneficial.**

We agree that this should be the case. The data used for the tables has two different sources with an overlapping period (1962-1979). The difference in the values is due to the extent of each period. For instance, in **Table 7** the values 3.25 and 3.75 Mw

are the class marks for the years 1978 and 1975, respectively. Meanwhile, in **Table 8** the corresponding completeness magnitude value for the period from 1962 to 1979 is 3.4 Mw and has been computed as the spatial average for the South-eastern Spain area using the data from *(González, 2017).* This procedure combined with an inequal development of the seismic network could explain the smooth changes in the completeness magnitude.

The completeness magnitude values used in this work are the following and now appear as a single table in the manuscript, note that the values from *Gaspar-Escribano et al. (2015)* corresponding to the years 1975 and 1978 have not been used as the data from *(González, 2017)* allows to further discretize the timeline:

| Years | 1048-1520 | 1521-1800 | 1801-1883 | 1884-1908 | 1909-1962 | 1963-1979 | 1979-1984 |
|---|---|---|---|---|---|---|---|
| Completeness magnitude [$M_w$] | 6.25 | 5.75 | 5.25 | 4.75 | 4.25 | 3.4 | 3.3 |

| Years | 1984-1992 | 1993-1998 | 1998-2002 | 2002-2010 | 2010-2013 | 2013-2023 |
|---|---|---|---|---|---|---|
| Completeness magnitude [$M_w$] | 3.0 | 2.9 | 2.3 | 2.1 | 1.9 | 1.8 |

The abrupt change between the period 1909-1962 and 1963-1979 is to be expected, as more development in the seismic network had been made in the mid-20th century than in the early 20th century given the political and historical context of Spain.

The paragraph before the table has been modified to explain the values used in this work.

*"Gaspar-Escribano et al. (2015) defined different threshold magnitudes for different regions around Spain. The class marks of these magnitude intervals for the zone of interest (South-eastern Spain) have been selected as the completeness magnitudes up until 1962. From 1962 on, the completeness magnitude has been computed by spatially averaging the gridded completeness magnitude results available from González (2017) over the area of study. The completeness magnitude values used in this work are presented in Table 7."*

*Table 7. Completeness magnitude for each period according to **Gaspar-Escribano et al. (2015)** in the top tabular and the spatially averaged completeness magnitude for each period using the results from **González (2017)** in the bottom tabular.*

| Completeness magnitude [$M_w$] | 6.25 | 5.75 | 5.25 | 4.75 | 4.25 |
|---|---|---|---|---|---|
| **From year** | 1048 | 1521 | 1801 | 1884 | 1909 |
| **to year** | 1520 | 1800 | 1883 | 1908 | 1962 |

| Completeness magnitude [$M_w$] | 3.4 | 3.3 | 3.0 | 2.9 | 2.3 | 2.1 | 1.9 | 1.8 |
|---|---|---|---|---|---|---|---|---|
| **From year** | 1963 | 1980 | 1985 | 1993 | 1999 | 2003 | 2011 | 2013 |
| **To year** | 1979 | 1984 | 1992 | 1998 | 2002 | 2010 | 2013 | 2023 |

**Model validation: Based on the results presented for the three models concerning both annual and monthly variations in the change of exceedance probability (as seen in Figures 12 and 13 and discussed in Lines 335-337), the authors assert that Model 1t outperforms the others. However, discerning significant differences is challenging, whether in the monthly variations of the relative change in annual probability of exceedance (Figure 12) or in the annual variations of the same (Figure 13). Moreover, I question the approach of basing model validation solely on 'greater changes before and after selected earthquakes' without incorporating statistical analyses. I believe that a more rigorous statistical evaluation is necessary to substantiate the claimed superiority of Model 1t.**

We agree that the discussion of the models' performance should be further explained. To do so, the figures of the results have been modified in order to be clearer and the results have commented in their respective sections and also in the conclusions.

[Figure]

*Figure 11. Relative change (RC) of the annual exceedance probability and corresponding uncertainty for Model 1t in Lorca, Murcia and Vera (from top to bottom). A zoom in on the mentioned increase in the RC during 1998 in Lorca's site appears in the upper left side of the graph.*

[Figure]

*Figure 12. Mean value of the relative change (RC) of the annual exceedance probability for models 1t, 2t and 3t in Lorca, Murcia and Vera (from top to bottom). At the right side, a zoom in on the peak due to Lorca earthquake.*

[Figure]

Figure 13. Annual and monthly variations of the relative change of the annual probability of exceedance for Model 1t in Lorca, Murcia and Vera (from top to bottom).

[Figure]

*Figure 14. Monthly variations of the relative change of the annual probability of exceedance for models 1t, 2t and 3t in Lorca, Murcia and Vera (from top to bottom). Two peaks have been selected for a zoom in detail (after Lorca earthquake in Lorca site and in December 2022 in Vera site). The earthquake that causes the peak in 1994 at Vera site has also been indicated.*

[Figure]

*Figure 15. Annual variations of the relative change of the annual probability of exceedance for models 1t, 2t and 3t in Lorca, Murcia and Vera (from top to bottom).*

*"Similar results are obtained for all three locations, although higher changes in the monthly variations (**Figure 14**) can be seen for Model 1t and Model 2t after Lorca's earthquake in Lorca site, and for Model 1t in December 2022 at Vera site. Overall, the monthly variations do not show changes preceding relevant earthquakes. One of the possible explanations is the lack of foreshocks in most of the main shocks. In Lorca earthquake, even though there was a 4.5 Mw earthquake almost two hours before the main-shock, the one-month increments on the computation process are not able to show any change in RC.*

*The annual variations on the other side (**Figure 15**), show periods of increased RC before some of the selected earthquakes. An example is seen in Lorca site where a 15% increase is seen before Mula earthquake from June 1998 (the earthquake occurred in February 1999). Another example can be seen in both Murcia and Lorca sites, where a 10% increase can be seen before Aledo earthquake from May 2004 until the earthquake occurrence in January 2005.*

*The most prominent increase on the annual variation occurs after Lorca earthquake (32.8%, 36.2% and 21% for models 1t, 2t and 3t)."*

In the conclusions, also:

**4.0 Conclusions**

*[...]*

*"Regarding which of the proposed models can be more effectively used to describe these changes, we have to consider several factors. One could be how close are the computed PGA values to the national seismic hazard maps for each country. In the case of Central Italy models 1t, 2t and 3t provide the following background PGA values: 340.61, 359.72 and 334.28 cm/s$^2$. The ESHM20 model (Danciu et al., 2021) computes 334.38 cm/s$^2$. The closest match would be Model 3t followed by Model 1t. However, by looking at Figure 5 and Figure 7, it can be seen that the Model 3t seems to be less responsive to the seismic activity than Model 1t, as the monthly and annual RC variations suggest (Model 1t monthly and annual variations are 4.5 times higher than Model 3t variations'). With this information Model 1t seems appropriate for the purpose of this work."*

**Figure 1 caption: Please provide clear definitions for each symbol and color used.**

The figure's caption has been updated to include clear definitions for each element of the map. Some changes in the figure have also been made, as some items where not able to be correctly read.

[Figure]

*Figure 2. Example of cluster identification in South-eastern Spain. **The cross markers represent the earthquakes and are coloured depending on the cluster they belong to. The events of the catalogue that do not belong to any cluster are***

*represented with grey circles. The fault traces are presented in red-coloured lines and the spatial grid cell's limits with green lines. A zoom in on Lorca's cluster is shown in the bottom right corner, where the events of the Lorca's series are plotted using purple circle symbols.*

**Lines 159: I am not quite sure if equation (7) is essential.**

We agree that it is indeed not essential, the equation 7 has been discarded, although the explanatory paragraph has been kept in order to support **Equation 8** (now **Equation 7**) explanation.

*"To do this, all the events belonging to each cluster are counted and define the clustering weight, $c_j$. If an event does not belong to any cluster (i.e., the cluster label for that event is set to zero) then $c_j$ equals 1. The weighted counts for each spatial grid cell are calculated as the summation of all the events over the different clusters (Eq. 7):"*

**Line 182: Revise 'time-dependent' as 'stability'.**

In order to be clearer about the b-value options the previous line has been rewritten as:

*"[...] a fixed and **constant** (time-independent) b-value assigned from the tectonic zones of each country [...]"*

We think it is better to keep the time-dependent adjective to the time-dependent model to emphasize the aim of the work, TDPSHA.

**Figure 3 caption: Is the tectonic b-value obtained by EHSM20 or by this study?**

Yes, the b-values are defined within the tectonic zonation from Danciu et al. (2021) -the EHSM20. In order to clarify this, the caption has been modified.

*"The map shows the tectonic zones (green lines) in Central Italy with their acronyms and tectonic b-values as computed in the EHSM20 (Danciu et al., 2021). The star marks the epicentre of L'Aquila earthquake (Table 2), and the red lines represent the fault traces."*

**Figure 4 caption: Please provide clear definitions for the dashed line.**

Figures from Figure 4 until the ones from the appendix have gotten their captions updated in order to have all the information needed for their analysis.

**Table 6: The effectiveness of the declustering approaches can be assessed with a confusion matrix, which provides not only the number of events but also the counts of true positives, true negatives, false positives, and false negatives.**

A figure presenting the confusion matrix for each of the declustering methods and the discussion, is now presented:

*"Considering that **Cabañas et al. (2011)** carried out a detailed study on the 2011 Lorca's earthquake seismic series, we have used their results to validate the best algorithm. According to them, the cluster corresponding to Lorca's series, from 11 May 2011 until 19 July 2011, is composed of 146 events (including the foreshock, the main shock and the aftershocks). In order to test the performance of the methods, the confusion matrices for each one have been computed. In the area of study, a total of 249 events have been recorded, which means a total of 103 background events should be identified. For this analysis, all the events classified in a cluster different from the one of Lorca series have been considered as background for simplicity. **Figure 9** shows that GK74 method is the most adequate (with a 94.43% mean for the metrics compared with the 92.88% for RJ and a 74.54% for A) and also the one that is able to identify more events belonging to Lorca's series."*

[Figure]

*Figure 9.* Confusion matrices for the tested declustering methods. Inside each square, the number of events (bold) and some metrics computed using the data are presented (NPV stands for Negative Predictive Value).

**Figures 14 and A1, Lines 370-372: Why is there an increase in the expected hazard and rate in Vera? Does this suggest that a large earthquake is anticipated in the future? An explanation based on the data and/or methodology used would be helpful.**

The sudden increase in December 2022, that adds up to the increasing RC tendency from 2012 onwards, can be related to the recent earthquake in a municipality 14 km from Vera site. This has been now addressed in the results' comments:

*"[...] Lastly, the peak in the annual and monthly variation at Vera in 2022 appears due to the seismic activity in Turre (a town 14 km south from Vera) where a 4.02 Mw earthquake struck on 31 December 2022."*

As for the increasing tendency itself, there is not enough data in order to formulate a hypothesis. It could be due to changes in the background seismicity (meaning a new update is due for the PGA background in Vera). In the last paragraph of the conclusions:

*"Finally, in the case of south-eastern Spain, the PSHA kept high in the region after the Mula earthquake and did not decrease until the occurrence of the Lorca earthquake. However, the continuous increase of the PSHA in Vera after the Lorca earthquake cannot be directly related to a potential upcoming earthquake similar to the one from Lorca. Therefore, more time and data are needed to confirm this."*

---

## Referee Report (RR1)

I am pleased to see the improvements made to this manuscript upon revision. The current version thoroughly details the seismic hazard approach developed and discusses the credibility of the method. However, based on their description, I am unclear why the authors have proposed a new smoothing method when other approaches with better performance are available. Therefore, I recommend further discussion on the credibility of this new smoothing approach, as detailed below.

Major concerns:

1D Gaussian function: I am puzzled as to why authors have proposed a new smoothing method based on a one-dimensional distance framework. This assumption might distort the distribution of seismic activity. Consider Figure 1d, for instance; the method presupposes a uniform seismic rate around the target site's periphery (full ring), even though there are no seismic sources to the southeast or northwest. Furthermore, there are already several established smoothing methods utilizing two-dimensional (e.g., Frankel, 1995; Woo, 1996) and even three-dimensional (e.g., Chan, 2016) distance frameworks. The rationale behind introducing this new approach remains unclear to me.

References:
Chan, C. H. (2016). Importance of three-dimensional grids and time-dependent factors for applications of earthquake forecasting models to subduction environments. Natural Hazards and Earth System Sciences, 16(9), 2177-2187.
Frankel, A. (1995). Mapping seismic hazard in the central and eastern United States. Seismological Research Letters, 66(4), 8-21.
Woo, G. (1996). Kernel estimation methods for seismic hazard area source modeling. Bulletin of the Seismological Society of America, 86(2), 353-362.

Discussion on credibility of the approach: I am pleased to see the discussion on the results of this time-dependent PSHA, highlighting the impact on seismic hazard from each event. At the same time, I would recommend further discussion on the credibility of this new smoothing approach. When introducing a new forecasting method, establishing its credibility is crucial, and retrospective validation could be an effective means to achieve this.

Validation of the declustering approaches: To validate the declustering approaches, the authors compared the results with the 2011 Lorca earthquake seismic series as defined by Cabañas et al. (2011), detailed in Lines 320-327. However, it is challenging to assert that this series, defined by a previous study, represents the

ground truth. In my view, the definition of an aftershock (how to determine if two events are related) is still contentious. I would appreciate some discussion on this topic in the manuscript if possible.

Minor Comments:
- b-value calculation (Line 227): Are there sufficient events to support the calculation of the b-value? Cases with an insufficient number of events (for instance, fewer than 100, as noted by Aki, 1965) could lead to greater uncertainties in the b-value. It would be beneficial to address this issue in the paper.

References:
Aki, K. (1965). Maximum likelihood estimate of b in the formula log N= a-bM and its confidence limits. Bull. Earthquake Res. Inst., Tokyo Univ., 43, 237-239.

- Table 9: Some parameters are also from Table 9?

Chung-Han Chan, National Central University, Taiwan, April, 2024.

---

## Referee Report (RR2)

*Referee Report of the article "Computing time-dependent activity rate using non-declustered and declustered catalogues. A first step towards time-dependent seismic hazard calculations for operational earthquake forecasting" (egusphere-2023-2818-manuscript-version2)*

The article is difficult to read. The structure of the material is chaotic. It is recommended to revise the presented material for better understanding of the readers. To work on the text, it may be useful for authors to eliminate the following shortcomings (the list is not exhaustive).

- The article discusses the uncertainty of epicentres, while at the same time seismic faults have no "width" but are defined by lines on the Earth's surface. This approach requires clarification in the text of the article

- Figure 1 - All four panels should be in the same scale. Figure 1c differs in scale from figures a, b and d. It is necessary to correct the parameter-values in the figure with the description in the text. For example, in the model description the parameter "mu" is defined as zero, whereas in Figure 1a it is given as 68.64. Explain the meaning of the colour code in Figure 1. To better understand the method of determining the parameter "mu" from fault locations, consider to add the additional lines to Figure 1. Swap figures 1d and 1c. Axis labelling should be added.

- Paragraph 2.2 lacks a description of the grid spacing used in the analysis. There should also be a discussion of the choice of grid spacing used.

- 188- there is no reference to the section describing the comparison of several declustering algorithms.Для рис 2 не указано какой алгоритм для выделения кластеров был использован.

- OpenQuake or Openquake use a single caption

- 299 – 304 The sudden mention of the NN method when discussing the results of the 3 already selected declustering methods is not clear.

- Fig 9 AF – is an abbreviation for Algorithm A? How do the data for Algorithm A from Fig. 9 and Table 6 agree, where 3394 events are listed for the Lorca series? The huge discrepancies between method A and the other two cluster extraction methods remain uncommented on in the article text.

- The data in Table 9 refers to Table 9. What is meant by this?

- For the Italian catalogue, calculations are from 2005. It is not clear why Table 3 data is given.

- For the catalogue of Spain, Calculations are made from 1990. It is not clear how the catalogue data from 1396 (and in Table 7 from 1048) are used in the calculations.
- For the neighbourhood of the L'Aquila earthquake and calculations in the vicinity of the epicentres of the three events in Spain it is not specified how the sizes of these neighbourhoods are chosen ? 200x200 cells - the regular grid cell sizes should be specified in km since the parameter of the epicentral uncertainty  is specified in kilometres.
- In Figure 14 and Figure 15, the differences in the models shown are indistinguishable. A different presentation of the material should be chosen to demonstrate the convergence (or difference).
- 429  "Model 1" - which model is this referring to?
- 441- 442 The sentence needs to be rewritten as PSHA - Probabilistic Seismic Hazard Analysis cannot be "high in the region..." or "continuous increase...".

---

## Referee Report (RR3)

I am pleased to observe significant improvements in the revised manuscript. I only have some technical concerns that I would like to address as follows:

Figure 2: Could you clarify the meanings of the crosses in various colors?

Regarding the Italy case study, multiple catalogs have been used, including HORUS (Line 256), EHSM20 (Figure 4 caption), and CPTI15 (Line 268). I recommend using a single catalog for consistency. If using multiple sources is necessary, please provide a justification for the selection of earthquake parameters from these diverse sources.

I eagerly anticipate the publication of this study!

Chung-Han Chan, National Central University, Taiwan, July, 2024.

---

## Author Response (AR2)

Dear Prof. Chan, thank you for your comments and corrections. Please, find in this document the response to the questions and concerns for the second version of the manuscript.

**Major concerns:**

1) **1D Gaussian function: I am puzzled as to why authors have proposed a new smoothing method based on a one-dimensional distance framework. This assumption might distort the distribution of seismic activity. Consider Figure 1d, for instance; the method presupposes a uniform seismic rate around the target site's periphery (full ring), even though there are no seismic sources to the southeast or northwest. Furthermore, there are already several established smoothing methods utilizing two- dimensional (e.g., Frankel, 1995; Woo, 1996) and even three-dimensional (e.g., Chan, 2016) distance frameworks. The rationale behind introducing this new approach remains unclear to me.**

**Response:**

We apologize for the confusing explanation; the Gaussian function in this work is a generalization of the one presented by Frankel (1995) with the difference that the main parameters are identified using different geophysical magnitudes, and it is not always centred in zero, hence it is not 1D. The presentation of the smoothing function has been changed in order to be clearer:

"*For this work, a modification of the kernel proposed by Frankel (1995) has been used to smooth the gridded seismicity (Eq. 6):*"

As for the Figure 1d the main consideration for the full ring to be created is that the distance from the spatial cell to both seismic sources is close, and also lower than the mean distance in between the sources in the region. For this to happen if the faults are the main seismic sources, it means that the part of the area of study being considered is densely populated with faults, in which case, considering the full ring could account for the seismicity distribution. The main difference with the classic Frankel (1995) approach would be that the weight is greater for the traces of the

faults and their surroundings. The Figure 1 has been modified to better suit the explanations in the section.

[Figure]

Figure 1. a) Smoothing function for $\mu = 0$. This can happen when either the distance is greater than $d_c$ or when the spatial cell is over the seismic source. b) Smoothing function for $\mu \neq 0$ i.e., when only one seismic source is present and with distance lesser than $d_c$. c) Smoothing function for $\mu \neq 0$ in the case that several seismic sources are surrounding the spatial grid cell at similar distances. d) Smoothing function for $\mu \neq 0$ when the spatial grid cell is near seismic sources with angular amplitude lesser than 180º. The blue lines show the fault traces. In this example $d_c$ equals 48 km. The stars mark the spatial grid cell considered in each case. A zoom in has been added in each panel to highlight the spatial distribution of smoothing kernel values.

2) **Discussion on credibility of the approach:** **I am pleased to see the discussion on the results of this time-dependent PSHA, highlighting the impact on seismic hazard from each event. At the same time, I would recommend further discussion on the credibility of this new smoothing approach. When introducing a new forecasting method, establishing its credibility is crucial, and retrospective validation could be an effective means to achieve this.**

We agree with the reviewer of the importance of the discussion on the credibility of the approach. That's why we have applied the smoothing methodology and the time-dependent PSHA to two different countries with different seismic activity. In both cases we have discussed how the exceedance probabilities change before given earthquakes and we have concluded that for high seismic activity regions (Italy) the methodology provides results that may be used for taking decisions before the main earthquake. However, for low seismic activity regions (Spain) the methodology is not so effective (at least with the data that we have). Here probably we will need more time to check if with a bigger earthquake the methodology behaves as in Italy.

In order not to extend the length of the manuscript too much we have clarified this discussion adding an introduction paragraph after section 3

"*3 Case studies*

*As explained before, the goal of our smoothing methodology is to test the viability of producing time-dependent seismic hazard results which may be used for taking decisions before the main earthquake. Therefore, now we will present and discuss the results obtained for two different regions with different seismic behaviour. Central Italy (high seismicity) and south-east Spain (low seismicity). We will check the if there are significant changes in the metrics before the occurrence of important earthquakes carrying out a retrospective validation of how useful the results are.*"

**Response:**

3) **Validation of the declustering approaches:** **To validate the declustering approaches, the authors compared the results with the 2011 Lorca earthquake seismic series as defined by Cabañas et al. (2011), detailed in Lines 320-327. However, it is challenging to assert that this series, defined by a previous study, represents the ground truth. In my view, the definition of an aftershock (how to determine if two events are related) is still contentious. I would appreciate some discussion on this topic in the manuscript if possible.**

**Response:**

We agree with the reviewer's comment. In fact, that's the reason why before explaining the comparison with Cabañas et al. (2011) we discussed the challenges on cluster definition using the works of Zaliapin and Ben-Zion (2020) and Anderson and Zaliapin (2023).

Anyway, we have also added your suggestion starting the sentence as follows:

*"In spite of the difficulties in defining the clusters, Cabañas et al. (2011) carried out a detailed study on the 2011 Lorca's earthquake seismic series. This study is the best definition at the moment so we will use it to validate the best algorithm."*
Minor Comments:

1) **b-value calculation (Line 227): Are there sufficient events to support the calculation of the b-value? Cases with an insufficient number of events (for instance, fewer than 100, as noted by Aki, 1965) could lead to greater uncertainties in the b-value. It would be beneficial to address this issue in the paper.**
   References:
   Aki, K. (1965). Maximum likelihood estimate of b in the formula log N= a-bM and its confidence limits. Bull. Earthquake Res. Inst., Tokyo Univ., 43, 237-239.

**Response:**

For the b-value computation the methodology explained in a previous work has been followed (Montiel-López et al. 2023). In this case, in order to ensure that enough events are considered in the analysis and that the b-value uncertainty remains low, a 1-year window has been selected.

Reference:

Montiel-López, D., Molina, S., Galiana-Merino, J. J., and Gómez, I. (2023). On the calculation of smoothing kernels for seismic parameter spatial mapping: methodology and examples, Natural Hazards and Earth System Sciences, 23, 91–106, https://doi.org/10.5194/nhess-23-91-2023

2) **Table 9: Some parameters are also from Table 9?**

**Response:**

We apologize for the mistake, the cross reference should have pointed to Table 8, this has been corrected so there is no circular reference.

First of all, we want to thank the referee_2 for the helpful comments and corrections that contribute to improve the quality of the manuscript. All of these questions have addressed in an orderly manner:

1) **The article discusses the uncertainty of epicentres, while at the same time seismic faults have no "width" but are defined by lines on the Earth's surface. This approach requires clarification in the text of the article.**

**Response:**

We agree that faults have been oversimplified by assuming they can be represented by their trace from QAFI database. We have decided to use their trace as most probable location near surface in order to define the maximum probability value for the seismicity smoothing model 1 (mu different from zero). The idea is to give more weight to seismic activity in the proximities of the faults. This has been done as for the two case studies the seismicity is shallow and mainly related to the faults.

Paragraph 1 from section 2.1.3 has been modified as follows:

"*In this section, some examples of how the smoothing kernel works are shown. In this case the seismic sources are faults in a shallow seismicity context, so the trace of such faults has been considered as the location with the maximum probability of having an earthquake. This approach has also been considered for the two case studies in this work. Three main scenarios have considered to showcase the smoothing kernel:*"

2) **Figure 1 - All four panels should be in the same scale. Figure 1c differs in scale from figures a, b and d. It is necessary to correct the parameter-values in the figure with the description in the text. For example, in the model description the parameter "mu" is defined as zero, whereas in Figure 1a it is given as 68.64. Explain the meaning of the colour code in Figure 1. To better understand the method of determining the parameter "mu" from fault locations, consider to add the additional lines to Figure 1. Swap figures 1d and 1c. Axis labelling should be added.**

**Response:**

Figure 1 has been corrected as there was an erroneous label in Figure 1a. The scales of each subfigure have been checked and a scale bar has been added to make all the subfigures comparable. The colour of each subfigure has been changed so there is only one common colour with its corresponding labelled colour bar. A different python library has been used to create the figures so there is regional context for each of the examples, in this way we hope it is not necessary to label the axis as they refer to the longitude and latitude.

[Figure]

Figure 2. a) Smoothing function for $\mu = 0$. This can happen when either the distance is greater than $d_c$ or when the spatial cell is over the seismic source. b) Smoothing function for $\mu \neq 0$ i.e., when only one seismic source is present and with distance lesser than $d_c$. c) Smoothing function for $\mu \neq 0$ in the case that several seismic sources are surrounding the spatial grid cell at similar distances. d) Smoothing function for $\mu \neq 0$ when the spatial grid cell is near seismic sources with angular amplitude lesser than 180º. The blue lines show the fault traces. In this example $d_c$ equals 48 km. The stars mark the spatial grid cell considered in each case.

3) **Paragraph 2.2 lacks a description of the grid spacing used in the analysis. There should also be a discussion of the choice of grid spacing used.**

**Response:**

We agree with the referee. The cell size of both case of studies is 0.015⁰x0.015⁰ which is equivalent to a ~1.5km$^2$ in surface. As for the choice of the cells the size we selected a size that could depict changes in the b-value and seismic activity rate spatial distribution and allowed for a reasonable computation time. Although for our study the deformation is not high (the longitude and latitude ranges are not wide) we prefer to use the ⁰x⁰ notation to avoid uncertainty. An example of the variations can be read in Wiemer and Wyss (2002) where the authors the typical values for nodal separation range between 0.5 km and 10 km. In the case of Spain, we computed a grid similar to the one in a previous work in the area of study (Montiel-López et al. 2023). For Central Italy, examples of grid sizes can be seen in Murru et al. (2016) with a 0.025⁰x0.025⁰ grid or Gulia and Wiemer (2019) with a 2-km spaced grid, so we compromised to a slightly higher definition (although the higher computation time) in order to use the same grid for both case studies. In general, for both cases the choice is also motivated by epicentre uncertainty for the most recent events in the catalogue (that belong to the studied periods).

The first paragraph in 2.2 has been modified to explain the reasoning behind the choice of the resolution of the grid.

"*First, the spatial grid is defined by creating a rectangle spanning the maximum and minimum longitudes and latitudes of the catalogue with the desired resolution. The choice of the resolution can be motivated by similar studies in comparable tectonic settings or the order of the epicentral uncertainty of the earthquakes in the catalogue. For this work, in the case of Spain the same resolution as in a previous work in the same area by Montiel-López et al. (2023) has been used. In the case of Italy, although Murru et al. (2016) use a 0.025⁰x0.025⁰ grid and Gulia and Wiemer (2019) use a 2-km spaced grid for Central Italy, we decided to use the same resolution for both case studies, a 0.015⁰x0.0.015⁰ grid.*"

Also in section 3.1.1, a correction has been made in the number of points as it was incorrect in the previous version of the manuscript:

*"A spatial cell grid of 0.015ºx0.015º spanning the above longitude and latitude ranges has been created (using 70756 points)."*

And in section 3.2.1:

*"A spatial grid of 0.015ºx0.015º covers the area of study (using 40401 points)."*

References:

Wiemer, S. & Wyss, M. (2002) Mapping spatial variability of the frequency-magnitude distribution of earthquakes. Adv. Geophys. 45, 259–302. doi: https://doi.org/10.1016/S0065-2687(02)80007-3

Murru, M., Taroni, M., Akinci, A. and Falcone, G. (2016) "What is the impact of the August 24, 2016 Amatrice earthquake on the seismic hazard assessment in central Italy?", Annals of Geophysics, 59. doi: https://doi.org/10.4401/ag-7209

Montiel-López, D., Molina, S., Galiana-Merino, J. J., and Gómez, I. (2023). On the calculation of smoothing kernels for seismic parameter spatial mapping: methodology and examples, Natural Hazards and Earth System Sciences, 23, 91–106, https://doi.org/10.5194/nhess-23-91-2023

Gulia L., Wiemer S. (2019) Real-Time Discrimination of Earthquake Foreshocks and Aftershocks. Nature,574, 193–199. doi: https://doi.org/10.1038/s41586-019-1606-4

4) **188- there is no reference to the section describing the comparison of several declustering algorithms.**

**Response:**

There was an error in the cross-reference pointing to section 3.2.1 that has been corrected:

*"In order to decide which algorithm performs best on the data, a comparison between the RJ, A, and GK74 declustering algorithms has been made using default parameters (see section 3.2.1)."*

5) OpenQuake or Openquake use a single caption.

**Response:**

All the instances in the text have been corrected to OpenQuake be coherent.

6) **299 – 304 The sudden mention of the NN method when discussing the results of the 3 already selected declustering methods is not clear.**

**Response:**

The mention to the works of the authors is related to the difficulties in assigning events to clusters and how it could affect to the seismicity hazard analysis, as this topic is discussed in the cited works. Nevertheless, the order has been changed so it comes before the table and is mainly focused on the results of the declustering.

*"Table 6 presents the number of clusters and the events in clusters for the whole seismic catalogue. The RJ algorithm classifies a total of 652 clusters in the catalogue while GK74 detects 1012 clusters. The A algorithm identifies 1245 clusters. As can be seen, despite the three methods relying on windows for their calculations, there are significant differences in the results, not only in the number of clusters but also in the number of events inside each cluster.*

*Zaliapin and Ben-Zion (2020) pointed out these problems with the identification of aftershocks and main shocks and proposed an algorithm to discriminate between background and clustered events by randomly thinning a complete catalogue by removing nearest-neighbour earthquakes. Moreover, Anderson and Zaliapin (2023) examine the effect on the hazard estimation when using different declustering thresholds. They conclude that hazard estimates are most sensitive to the catalogue thinning near the aftershock zone, and less sensitive elsewhere.*

*In spite of the difficulties in defining the clusters, Cabañas et al. (2011) carried out a detailed study on the 2011 Lorca's earthquake seismic series. This study is the best definition at the moment so we will use it to validate the best algorithm. They identified 146 events (including the foreshock, the main shock and the aftershocks) that belong to Lorca's series, from 11 May 2011 until 19 July 2011. With this information, in order to test the performance of the declustering methods, the confusion matrices for each one have been computed. In the area of study, a total of 249 events have been recorded, which means a total of 103 background events should be identified. For this analysis, all the events classified in a cluster different from the one of Lorca series have been considered as background for simplicity.*

*Figure 9 shows that GK74 method is the most adequate (with a 94.43% mean for the metrics compared with the 92.88% for RJ and a 78.79% for A) and also the one that is able to identify more events belonging to Lorca's series."*

Reference:

Cabañas, L., Carreño, E., Izquierdo, A., Martínez, J. M., Capote, R., Martínez-Díaz, J., Benito, B., Gaspar-Escribano, J., Rivas-Medina, A., García-Mayordomo, J., Pérez, R., Rodríguez-Pascua, M. A., and Murphy, P. (2011) Informe del sismo de Lorca del 11 de mayo de 2011 (in spanish), https://digital.csic.es/handle/10261/62381

7) **Fig 9 AF – is an abbreviation for Algorithm A? How do the data for Algorithm A from Fig. 9 and Table 6 agree, where 3394 events are listed for the Lorca series? The huge discrepancies between method A and the other two cluster extraction methods remain uncommented on in the article text.**

**Response:**

There is a typo in the figure, as the label should be A, not AF, which is referring to the Afteran's algorithm. The figure has been corrected. Regarding the discrepancy between the Fig.9 and Table 6 we have revised the results of the declustering and there was an error regarding the A algorithm. A correction has been made in the Figure 9, in Table 6 and in the paragraph that accompanies the results:

| Algorithm | RJ | A | GK74 |
|---|---|---|---|
| **Number of clusters** | 652 | 1245 | 1012 |
| **Events in clusters** | 7143 | 10167 | 7552 |
| **Events inside Lorca's series** | 123 | 196 | 136 |

[Figure]

*"Table 6 presents the number of clusters and the events in clusters for the whole seismic catalogue.  The RJ algorithm classifies a total of 652 clusters in the catalogue while GK74 detects 1012 clusters. The A algorithm identifies **1245** clusters. As can be seen, despite the three methods relying on windows for their calculations, there are significant differences in the results, not only in the number of clusters but also in the number of events inside each cluster.*

*[…]*

*In spite of the difficulties in defining the clusters, Cabañas et al. (2011) carried out a detailed study on the 2011 Lorca's earthquake seismic series. This study is the best definition at the moment so we will use it to validate the best algorithm. They identified 146 events (including the foreshock, the main shock and the aftershocks) that belong to Lorca's series, from 11 May 2011 until 19 July 2011. With this information, in order to test the performance of the declustering methods, the confusion matrices for each one have been computed. In the area of study, a total of 249 events have been recorded, which means a total of 103 background events should be identified. For this analysis, all the events classified in a cluster different from the one of Lorca series have been considered as background for simplicity. Figure 9 shows that GK74 method is the most adequate (with a 94.43% mean for the metrics compared with the 92.88% for RJ and a **78.79%** for A) and also the one that is able to identify more events belonging to Lorca's series."*

**8)  The data in Table 9 refers to Table 9. What is meant by this?**

**Response:**

There was an erroneous cross-reference to Table 8 that has been corrected.

**9) For the Italian catalogue, calculations are from 2005. It is not clear why Table 3 data is given.**

**Response:**

We agree that this should be clarified. This question is related with the next one. In order to compute the probability of exceedance for a certain strong motion, we consider a background value. For this reason, the catalogue used to compute the background ends in the start of 2007 for Central Italy (Figure 5), and the start of 1990 for Spain (related figures). Hence the need of the different completeness magnitudes shown in the Table 3 and Table 7, as the data prior 2007 and 1990 constitutes the background PGA the changes about which we are computing.

**10) For the catalogue of Spain, calculations are made from 1990. It is not clear how the catalogue data from 1396 (and in Table 7 from 1048) are used in the calculations.**

**Response:**

This is explained in the previous answer to the question 9).

**11) For the neighbourhood of the L'Aquila earthquake and calculations in the vicinity of the epicentres of the three events in Spain it is not specified how the sizes of these neighbourhoods are chosen ? 200x200 cells - the regular grid cell sizes should be specified in km since the parameter of the epicentral uncertainty is specified in kilometres.**

**Response:**

Typically, PSHA studies require 300 km of area around the point of interest in order to avoid lack of data and border effects. For the selected point, the rest of the points in the grid are used as point sources that contribute towards the seismic hazard. Nonetheless, for the selected GMPs the distance from this point to the rest are computed and their contribution is weighted accordingly, so the closer points are the ones that contribute the most. There is no preselection of the neighbourhood,

the function that computes the median spectral acceleration already depends on the distance, an example of the functional form can be seen in Akkar et al. (2013).

Reference:

Akkar, S., Sandikkaya, M. A., Bommer J. J. (2014) Empirical Ground-Motion Models for Point- and Extended- Source Crustal Earthquake Scenarios in Europe and the Middle East, Bulletin of Earthquake Engineering (2014), 12(1): 359 - 387

12) **In Figure 14 and Figure 15, the differences in the models shown are indistinguishable. A different presentation of the material should be chosen to demonstrate the convergence (or difference).**

**Response:**

We agree that the there are no distinguishable differences in Figure 14, for the given period, hence the need of defining other metrics that can be useful in low-to-moderate seismicity settings. As for the Figure 15, there is important differences in the models after Lorca's earthquake in 2011, meaning these selected metrics could be useful in the context of moderate earthquakes (as is the case of Lorca's series with 2 earthquakes with magnitude greater than 4.5).

Some modifications in the paragraphs that accompany Figure 14 and Figure 15 have been made:

"*Similar results are obtained for all three locations regarding the monthly variations (Figure 14). Overall, the monthly variations do not show changes preceding relevant earthquakes for this case study. One of the possible explanations is the lack of foreshocks in most of the main shocks. In Lorca earthquake, even though there was a 4.5 Mw earthquake almost two hours before the main-shock, the one-month increments on the computation process are not able to show any change in RC.*

*The annual variations (Figure 15), on the other hand, show periods of increased RC before some of the selected earthquakes. An example is seen in Lorca site where a 15% increase is seen before Mula earthquake from June 1998 (the earthquake occurred in February 1999). Another example can be seen in both Murcia and Lorca sites, where a 10% increase can be seen before Aledo earthquake from May 2004 until the earthquake occurrence in January 2005. For this metric, differences*

*between the three models can be seen. For instance, Model 1t and Model 2t show greater changes after Lorca's earthquake in Lorca's site."*

**13) 429 "Model 1" - which model is this referring to?**

**Response:**

Model 1 has been corrected to Model 1t to be coherent with this manuscript's notation. This correction has also been applied to the labels in Figure 14 and Figure 15.

**14) 441- 442 The sentence needs to be rewritten as PSHA - Probabilistic Seismic Hazard Analysis cannot be "high in the region..." or "continuous increase...".**

**Response:**

We agree that PSHA cannot be addressed in terms used in the aforementioned sentences. The text has been modified because it referred to the change in the probability of exceedance:

*"Finally, in the case of south-eastern Spain, the relative change in the annual probability of exceedance kept high in the region after Mula earthquake and did not decrease until the occurrence of the Lorca earthquake. However, the continuous increase in this parameter in Vera after the Lorca earthquake cannot be directly related to a potential upcoming earthquake similar to the one from Lorca. Therefore, more time and data are needed to confirm this."*

First of all, we thank referee_3 for the comments and suggestions. Please, find in the document in an orderly manner the answers to the comments:

**- The main message of the paper is unclear, if not wrong. From one side the authors seem to focus on classical long-tern PSHA that is used for building code purposes (return periods of 475 years, which corresponds to exceedance probability of 10%in50 years). Of course, PSHA can be rescaled to shorter forecasting time windows, but, if it is used for land use planning the time windows remain long. Conversely, they also address the short time scales of the operational earthquake forecasting (OEF) whose forecasts are updated daily (if not more frequently during a sequence) because they are used to manage earthquake sequences. In other words, the updating time window and the forecasting time window are both very important for practical applications and should be coherent with the use of the models (land use planning or emergency management).**

We agree with the concepts of PSHA and OEF you have explained, however the main message of the paper has been misunderstood by the reviewer. We proposed a smoothing methodology that can be used to compute first a long-term background PSHA which will be later compared with a short-term time dependent SHA so we will investigate metrics that can be used to take decisions before the main earthquake (similarly to OEF). We think that the changes introduced in the manuscript, also with the help of the other reviewers, have helped to clarify this.

**- Many recent papers, that are not cited here, have already addressed the inclusion of earthquake clustering in PSHA. Besides quoting the relevant literature on this point, it would be interesting to explain why the authors think that their procedure would be better.**

We do not pretend to conclude which is the best procedure in the definition of earthquake clustering but to compare several procedures and choose the one who are closer to the better definition of clustering. That's the reason why before explaining the comparison with Cabañas et al. (2011) we discussed the challenges on cluster definition using the works of Zaliapin and Ben-Zion (2020) and Anderson and Zaliapin (2023).

Anyway, we have improved the paragraphs as follows:

"*In spite of the difficulties in defining the clusters, Cabañas et al. (2011) carried out a detailed study on the 2011 Lorca's earthquake seismic series. This study is the best definition at the moment so we will use it to validate the best algorithm.*"

- **The method proposed by the authors estimates the earthquake rate and b-value from a short-time window (for instance during a foreshock sequence) and then it extrapolates them to the next decades (classical time scale of long-term PSHA). This is clearly wrong, because both parameters have a large variability on the long-term forecasting time window of classical PSHA (decades) and they cannot be considered stationary in such a time window.**

**Current methods to incorporate earthquake clusters in PSHA consider that the time evolution of the earthquake clustering rapidly decays with time and it is much shorter than the forecasting time window.**

The reviewer has misunderstood the concept. As explained before compute first a long-term background PSHA which will be later compared with a short-term time dependent SHA. For the short-term SHA the a and b parameters have been obtained using a time window of enough time to assure we will be able to get reliable results.

- **As said before, many recent PSHA initiatives account for aftershocks and foreshocks in different ways. Despite the differences among these procedures, all of them agree on the fact that declustering has to be necessarily applied for reducing the spatial bias. This is not done in this paper. In essence, if no declustering is applied, it is tacitly assumed that earthquake clusters that will occur in the next future will have preferentially the same locations of the past clusters. This has no scientific basis according to the present state of knowledge (earthquake clusters may happen everywhere).**

We agree with the reviewer. That's why our smoothing methodology investigates different approaches to obtain the gridded seismic activity rate.

- **Very often the b-value varies with time due to catalog incompleteness, which is pervasive after a major shock and it can then induce to string bias in hazard calculations. It is not clear if the authors address this important issue properly.**

We agree with the reviewer. That's why we have addressed the importance of a using different magnitudes of completeness when obtaining a and b.

---

## Author Response (AR3)

Dear Referee#1 thank you for your comments and suggestions as they helped us to improve the manuscript and clarify some important aspects. In the following pages we address the concerns raised in your review and try to answer them accordingly.

1) **In the revised version of the manuscript, the authors have corrected numerous typographical errors and inconsistencies. However, it remains challenging for both the reviewer and potential readers to assess the accuracy of the calculations and results presented. Notably, data from the catalog used for the territory south and southeast of Spain is not readily available in the public domain. For the Italian catalog, even a superficial assessment of the data presented by the authors reveals discrepancies between the seismic parameters reported in the article and those in the HORUS catalog (https://horus.bo.ingv.it/). For instance, data about the test event in L'Aquila (Table 2) do not match the corresponding entry in the HORUS catalog.**

| Name | Latitude(ºN) | Longitude(ºE) | Depth (km) | Mw |
|---|---|---|---|---|
| L'Aquila (former) | 42.334 | 13.334 | 10 | 6.1 |
| L'Aquila (HORUS) | 42.342 | 13.380 | 8.3 | 6.29 |

**«255 It has a total of 49112 events with maximum depth of 30 km and maximum magnitude of 6.1 Mw». In the HORUS catalog, from 1960 to 2012, there are 99,266 events with depths ranging from 0 to 100 km. The strongest event in the zone selected for analysis is the Irpinia earthquake on November 23, 1980, which has a magnitude of 6.81 according to the HORUS catalog. The discrepancy raises concerns about the reliability of other data and calculations presented in the manuscript. Given these issues, the reviewer recommends a thorough revision of the source data and calculations. The manuscript should be significantly reworked, with an emphasis on ensuring that the results can be reproduced and evaluated by interested readers for validity and**

**accuracy. This would not only enhance the credibility of the study but also its utility to the scientific community.**

**Response:**

We agree with the referee, the above-mentioned issue induces confusion about the data source. The case of the Table 2 regarding L'Aquila earthquake comes from a mistake, as the table contained at first (not anymore after this revision) a placeholder information about the earthquake obtained from a source different from the HORUS catalogue. This has been changed as all the information in tables should come from the same source used in the computation.

The following fields in Table 2 have been updated:

| Name | Latitude(ºN) | Longitude(ºE) | Depth (km) | Mw |
|------|--------------|---------------|------------|------|
| L'Aquila (AB) | 42.342 | 13.38 | 8.3 | 6.29 |

The epicentral distance to L'Aquila remains unchanged as the values of longitude and latitude used for the computation of said parameter are the ones from the HORUS catalogue (which now appear in the table).

Table 5 has also been updated to show two decimals, in case there are such, for the Mw:

So, for Mula the Mw is now 5.86 instead of 5.9. The rest of the events only had at most 1 decimal place, so they remain unchanged.

Regarding the number of events that appear in the text, the second paragraph of the subsection 3.1.1 in the manuscript has been modified with the corresponding explanation:

"*Figure 4 shows the location of the area of study (a rectangle with longitudes from 11.39**4** to 15.3**91**º E and latitudes from 40.3**59** to 44.353º N) and the tectonic zones and main faults **as defined by** Danciu et al. (2021) **to compute the European Seismic Hazard Map**. For this area, the Italian HORUS catalogue (Lolli et al., 2020) has been used as it has been homogenised and comprises events from 1960 to 20**23** (in order to study this particular seismic series). **A spatial filtering process has been applied to the catalogue to extract the events within the above-mentioned***

***area, so 49112 events with maximum depth of 30 km and maximum magnitude of 6.81 Mw remain."***

There were some rounding errors that affected the values of the specified bounding rectangle area that have been also corrected to reflect the maximum and minimum coordinates in the catalogue. The previous version of the manuscript contained these values "*11.392 to 15.372º E and latitudes from 40.374 to 44.354º N*", whereas the new version contains these "*11.39**4** to 15.3**91**º E and latitudes from 40.3**59** to 44.353º N*". This difference could be either due to rounding error while using Python to compute these coordinates or a mistake writing the manuscript. In order to be coherent, the maximum and minimum values have been checked with those that appear in the HORUS catalogue. The magnitude range has also been revised, as the maximum magnitude is indeed 6.81 Mw from the Irpina earthquake. The year range of the catalogue has also been modified from 1960 to 2023 (the previous version stated 1960 to 2012 and was a mistake) as it is a description of the full catalogue for this area.

In this sense the catalogues for both locations have been uploaded in an online GitHub repository so they can be checked. It will also be added as either a supplementary material or a link inside the manuscript. URL: https://doi.org/10.5281/zenodo.13691624

"*Data availability. The catalogues used in this work can be found in the following repository: https://doi.org/10.5281/zenodo.13691624*"

Dear Professor Chan, thank you for your comments and questions as they help us to improve the manuscript. In the following pages, the questions and comments are addressed in an orderly manner.

1) **Figure 2: Could you clarify the meanings of the crosses in various colours?**

Response:

The events that belong to the same cluster have the same colour, as this colour is assigned using the cluster numerical ID field. Figure 2 caption has been modified to have a clearer explanation as the phrasing was confusing. In this later version it can be read as:

"*Figure 2. Example of cluster identification in South-eastern Spain. The cross markers represent events in the catalogue which belong to a cluster while the grey circles are the events not belonging to any cluster. All the cross markers with the same colour indicate they are in the same cluster. The fault traces are presented in red-coloured lines and the spatial grid cell's limits with green lines. A zoom in on Lorca's cluster is shown in the bottom right corner, where the events of the Lorca's series are plotted using purple circles, the size of which depends on the magnitude of the event.* "

2) **Regarding the Italy case study, multiple catalogs have been used, including HORUS (Line 256), EHSM20 (Figure 4 caption), and CPTI15 (Line 268). I recommend using a single catalog for consistency. If using multiple sources is necessary, please provide a justification for the selection of earthquake parameters from these diverse sources.**

We agree that this should be addressed. We want to clarify in the first place that only one catalogue has been used in the computations, that is, the HORUS catalogue (Lolli et al., 2020).

The reference ESHM20 (Figure 4 caption) is not related to the use of a different catalogue but to the seismic zonation proposed in the current European seismic Hazard Map and how they divide the region in zones with different tectonic values

(b-values). We changed some mentions in the manuscript to avoid misunderstandings:

"*Figure 4. The map shows the tectonic zones (green lines) in Central Italy with their acronyms and tectonic b-values as can be found in Danciu et al. (2021)* *.*"

*Line 254: "...... and the tectonic zones and main faults as defined by Danciu et al. (2021) to compute the European Seismic Hazard Map."*

The reference CPTI15 (Line 268) was included to justify the smoothing value used for the spatial uncertainty of the epicentral location of the earthquakes in Italy since the HORUS catalogue has no information about the location uncertainty. The HORUS catalogue is a homogenized instrumental catalog based on the hypocentral location of earthquakes compiled from the Italian Seismological Instrumental and parametric Database (ISIDe) and therefore their spatial uncertainty can be deduced from Rovida et al. (2022).

Since the smoothing value (sigma) used in our computations and based on the location uncertainty aims to account for the physical variability in the location of the earthquakes, we have tested two models with different uncertainty values to showcase the variability in the results as a consequence of increasing or decreasing the uncertainty of the epicenter location. Additionally, the work from Scudero et al. (2021) gives insight on the variation of the horizontal error (ERH) in Italy as well as a range of mean values for different revision processes on the data (2.2, 3.3 and 13.1 km). This range seems coherent with the uncertainties inside CPTI15 catalogue (Rovida et al., 2022, 2020).

Therefore, a minimum value of 6 km, in agreement with the previous explanation, and a maximum value of 30 km following the work of Taroni et al. (2021) has been chosen to characterize our spatial uncertainty.

We have removed the reference to CPTI15 to avoid misunderstanding but kept the reference to Rovida et al. (2022).

We have modified the manuscript to reflect this process and the reasoning behind it to clarify it to the reader:

"In order to characterise the spatial uncertainty on the epicentral location, two different possibilities have been defined. A minimum value of 6 km, in agreement with the mean epicentral uncertainty during the last decades of the instrumental seismicity detected by the Italian Seismic Network (Scudero et al., 2021; Rovida et al., 2022) and a maximum value of 30 km following the work of Taroni et al. (2021).

New references:

Scudero, S., Marcocci, C. & D'Alessandro, A. Insights on the Italian Seismic Network from location uncertainties. J Seismol 25, 1061–1076 (2021). https://doi.org/10.1007/s10950-021-10011-6

---

## Author Response (AR4)

Dear Veronica Pazzi,

We have made all the proposed changes in the manuscript.

We added missing citations for the catalogue's, as well as the reference to the data availability section (in which the references to the raw catalogues have been also cited).

The Figure 2 caption has been changed.

The explanation on the smoothing kernel values for Italy have been expanded in order to be similar to the one given in the referee answer:

"*Since the σparameter used in the smoothing kernel computations and based on the location uncertainty aims to account for the physical variability in the location of the earthquakes, three models with different uncertainty values have been tested to showcase the variability in the results as a consequence of increasing or decreasing the uncertainty of the epicentre location. In order to obtain such values, the work from (Scudero et al., 2021) gives insight on the variation of the horizontal error (ERH) in Italy as well as a range of mean values for different revision processes on the data (2.2, 3.3 and 13.1 km). Given that the HORUS (Lolli et al., 2020) catalogue, used in this work, has no information on the ERH, but the locations of the events are obtained through the ISIDe database, their spatial uncertainty can be deduced from the CPTI15 catalogue (Rovida et al., 2020, 2022).*

*The aforementioned range of mean values for the ERH is coherent with the mean spatial uncertainty obtained from the CPTI15 catalogue.*

*Therefore, a minimum value of 6 km, in agreement with the previous explanation, and a maximum value of 30 km, following the work of Taroni et al. (2021) has been chosen to characterise the spatial uncertainty. The three models proposed for the seismic activity smoothing are presented in Table 4.*"

The new version of the manuscript is sent along with the pdf with the changes.

Best regards,
The authors